# Pareto Domain Adaptation

**Fangrui Lv,**[1,*]    **Jian Liang,**[2,*]    **Kaixiong Gong,**[1]    **Shuang Li,**[1,†]
**Chi Harold Liu,**[1]    **Han Li,**[2]    **Di Liu,**[2]    **Guoren Wang**[1]

[1] Beijing Institute of Technology, China    [2]Alibaba Group, China

[1] {fangruilv,kxgong,shuangli,wanggrbit}@bit.edu.cn, liuchi02@gmail.com
[2] {xuelang.lj,lihan.lh,wendi.ld}@alibaba-inc.com

## Abstract

Domain adaptation (DA) attempts to transfer the knowledge from a labeled source domain to an unlabeled target domain that follows different distribution from the source. To achieve this, DA methods include a source classification objective $\mathcal{L}_S$ to extract the source knowledge and a domain alignment objective $\mathcal{L}_D$ to diminish the domain shift, ensuring knowledge transfer. Typically, former DA methods adopt some weight hyper-parameters to linearly combine the training objectives to form an overall objective $\mathcal{L}$. However, the gradient directions of these objectives may conflict with each other due to domain shift. Under such circumstances, the linear optimization scheme might decrease the overall objective value at the expense of damaging one of the training objectives, leading to restricted solutions. In this paper, we rethink the optimization scheme for DA from a gradient-based perspective. We propose a Pareto Domain Adaptation (ParetoDA) approach to control the overall optimization direction, aiming to cooperatively optimize all training objectives. Specifically, to reach a desirable solution on the target domain, we design a surrogate loss mimicking target classification. To improve target-prediction accuracy to support the mimicking, we propose a target-prediction refining mechanism which exploits domain labels via Bayes' theorem. On the other hand, since prior knowledge of weighting schemes for objectives is often unavailable to guide optimization to approach the optimal solution on the target domain, we propose a dynamic preference mechanism to dynamically guide our cooperative optimization by the gradient of the surrogate loss on a held-out unlabeled target dataset. Our theoretical analyses show that the held-out data can guide but will not be over-fitted by the optimization. Extensive experiments on image classification and semantic segmentation benchmarks demonstrate the effectiveness of ParetoDA. Our code is available at *https://github.com/BIT-DA/ParetoDA*.

## 1   Introduction

Domain adaptation (DA) is a well-established paradigm for learning a model on an unlabeled target domain with the assistance of a labeled related source domain, which has attracted a surge of interests in the machine learning community [34, 33, 13]. By bridging the domain gap on the premise of ensuring the performance of source classification task, DA can adapt the model learned from the source domain to the target in the presence of data bias, which solves the dilemma of label scarcity in many real-world applications [55, 47, 11, 45, 3, 57]. Extensive DA approaches have been proposed in recent years, achieving great performances in many areas such as image classification [11, 27, 56], semantic segmentation [49, 52, 16], and object detection [39, 43, 54].

---

∗ Equal contributions from both authors.
† Corresponding author.

35th Conference on Neural Information Processing Systems (NeurIPS 2021).

Most DA methods are devoted to aligning the distributions across domains by explicitly minimizing some discrepancy metrics [26, 28, 56] or adopting adversarial learning [11, 27, 41]. However, since the domain alignment objective also depends on the target domain that deviates from the source, the gradient direction of it may conflict with that of classification objective on the source domain. Thus, when the optimization proceeds to a certain stage, one objective will deteriorate if further improving the other one. Under this circumstance, no solution can reach the optimal of each objective at the same time. We can only obtain a set of so-called Pareto optimal solutions instead, where we cannot further decrease all objectives simultaneously [10]. All Pareto optimal solutions in loss space compose Pareto front as shown in Fig. 1.

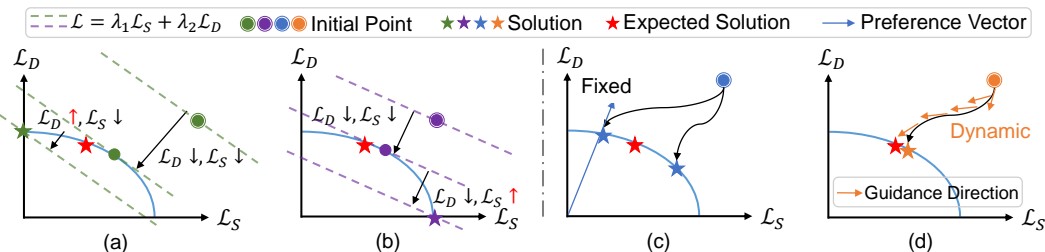

Figure 1: Illustration of different optimization schemes. In each panel, the blue curve is the Pareto front where the region underneath is unaccessible. (a)-(b): Linear scheme that adopts weight hyper-parameters to unify the objectives. The green and purple dash lines represent different hyper-parameters. (c): Previous gradient-based scheme, which reaches Pareto optimal solutions with or without the guidance of preference vector. (d): Our ParetoDA that dynamically guides the optimization by the gradient of the target-classification-mimicking loss on a held-out unlabeled target dataset, approaching the expected solution that minimizes the target classification loss.

This dilemma obviously has been overlooked by most former DA methods, which generally adopt some empirical weight hyper-parameters to linearly combine the objectives, constructing an overall objective. This linear weighting scheme has two major concerns. First, as pointed out in [4], it can only obtain solutions on the convex part of the Pareto front of the objectives, while cannot reach the non-convex part, as shown in Fig. 1 (a)-(b). That is because it only considers reducing the overall objective value, which might damage some objectives during optimization (see the red up-arrow). Unfortunately, the Pareto fronts of DA methods are often non-convex, due to loss conflict caused by domain shift. See examples shown in Fig. 2. Second, one could hardly reach the desired solution that performs best on target domain by tuning the weight hyper-parameters, since the obtained solution will deviate severely when the trade-offs slightly change (see the slopes of the dash lines in Fig. 1 (a)-(b) for reference).

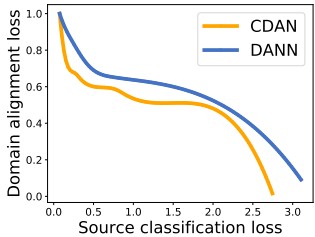

Figure 2: The non-convex Pareto fronts of DANN [11] and CDAN [27] on task W→A (Office-31).

To remedy these issues, we rethink the optimization scheme for DA from a gradient-based perspective. By weighting objectives to control the overall optimization direction so that no objective deteriorates, one can reach a Pareto optimal solution on the Pareto front [1, 23, 30, 32]. Moreover, [30] precisely sets objective weights according to a prefixed preference vector (see Fig. 1 (c)). However, neither of them suits DA, because 1) in DA, the goal is to minimize the target classification loss $\mathcal{L}_T^*$, but neither $\mathcal{L}_S$ or $\mathcal{L}_D$ directly corresponds to this goal; 2) the target domain has no label to construct a classification loss for the goal; 3) prior knowledge of weighting schemes for objectives is often unavailable to guide optimization to approach the optimal solution that minimizes $\mathcal{L}_T^*$.

In this paper, we propose a Pareto Domain Adaptation (ParetoDA) approach, which cooperatively optimizes training objectives and is specifically designed for DA. Specifically, to minimize $\mathcal{L}_T^*$, we design a target-classification-mimicking (TCM) loss $\mathcal{L}_T$ by the mutual information [17] leveraging the target predictions. In this loss, target predictions act as (soft) labels for themselves. Therefore, the accuracy of the predictions is important to support the mimicking. To refine the predictions by maximally exploiting the information at hand, we propose a *target prediction refining mechanism* which models the domain label as conditional information into the prediction probability by Bayes' theorem. Intuitively, we ensembles a class-wise domain discriminator (trained with the domain labels)

as another classifier—*only when the class condition is correct, the discriminator can correctly predict the domain*. On the other hand, to weight $\mathcal{L}_T, \mathcal{L}_S$ and $\mathcal{L}_D$ to guide the cooperative optimization towards minimizing $\mathcal{L}_T^*$, we propose a *dynamic preference mechanism* to model the gradient of $\mathcal{L}_T$ on a held-out unlabeled target dataset as the guidance. We evaluate the performance on the held-out data because 1) it suggests generalization performance, and 2) it is more stable and convincible than training data. One may be concerned that involving the gradient of the held-out data in training may eventually cause over-fitting of the data. Fortunately, our theoretical analyses show that the held-out data can guide but will not be over-fitted by the optimization of our method. Our contributions are:

- We rethink the optimization scheme in existing DA methods, which can only reach restricted solutions if the optimization objectives conflict with each other. And existing gradient-based strategies do not suit DA.

- We propose a Pareto Domain Adaptation method to obtain the desired Pareto optimal solution learned and dynamically guided (using the held-out unlabeled target data) by the designed surrogate objective function that mimics target classification. To better support the mimicking, we also propose a target prediction refining mechanism. As a general technique, ParetoDA can be plugged into various DA methods and enhance their performance.

- We evaluate ParetoDA on two DA applications: image classification and semantic segmentation. Experimental results demonstrate the effectiveness of ParetoDA.

## 2 Background

Consider $m$ objectives, each with a non-negative loss function $\mathcal{L}_i(\boldsymbol{\theta}) : \mathbb{R}^d \to \mathbb{R}_+, i \in [m] = \{1, 2, ..., m\}$, where $\boldsymbol{\theta}$ is the shared parameters of the objectives to be optimized. There may exist no single solution that can reach the optimal solution of each objective at the same time due to they may conflict with each other. What we can obtain instead is a set of so-called Pareto optimal solutions, the related definitions are as follows [58]:

**Pareto dominance.** Suppose two solutions $\boldsymbol{\theta}_1, \boldsymbol{\theta}_2$ in $\mathbb{R}^d$, we define $\boldsymbol{\theta}_1 \prec \boldsymbol{\theta}_2$ if $\mathcal{L}_i(\boldsymbol{\theta}_1) \leq \mathcal{L}_i(\boldsymbol{\theta}_2), \forall i \in [m]$ and $\mathcal{L}_i(\boldsymbol{\theta}_1) < \mathcal{L}_i(\boldsymbol{\theta}_2), \exists i \in [m]$. We say that $\boldsymbol{\theta}_1$ dominates $\boldsymbol{\theta}_2$ iff $\boldsymbol{\theta}_1 \prec \boldsymbol{\theta}_2$. Note that $\boldsymbol{\theta}_1 \nprec \boldsymbol{\theta}_2$ if $\boldsymbol{\theta}_2$ is not dominated by $\boldsymbol{\theta}_1$.

**Pareto optimality.** If a solution $\boldsymbol{\theta}_1$ dominates $\boldsymbol{\theta}_2$, then $\boldsymbol{\theta}_1$ is clearly preferable as it perform better or equally on each objective. A solution $\boldsymbol{\theta}^*$ is Pareto optimal if it is not dominated by any other solutions. That is, $\boldsymbol{\theta} \nprec \boldsymbol{\theta}^*, \forall \boldsymbol{\theta} \in \mathbb{R}^d - \{\boldsymbol{\theta}^*\}$.

**Pareto front.** The set of all Pareto optimal solutions in loss space is Pareto front, where each point represents a unique solution.

**Preference vector.** A Pareto optimal solution can be viewed as an intersection of the Pareto front with a specific direction in loss space (a ray). We refer to this direction as the preference vector.

In gradient-based optimization, we can reach a Pareto optimal solution by starting from an arbitrary initialization and iteratively finding the next solution $\boldsymbol{\theta}^{t+1}$ that dominates the previous one $\boldsymbol{\theta}^t$, i.e., $\boldsymbol{\theta}^{t+1} = \boldsymbol{\theta}^t - \eta \boldsymbol{d}_{des}$, where $\eta$ is the step size and $\boldsymbol{d}_{des}$ is the optimization direction. Let $\boldsymbol{g}_i = \nabla_{\boldsymbol{\theta}} \mathcal{L}_i$ be the gradient of the $i$-th objective function at $\boldsymbol{\theta}$, the descent direction $\boldsymbol{d}_{des}$ is given by $\boldsymbol{d}_{des}^T \boldsymbol{g}_i \geq 0, \forall i \in [m]$. In other words, the descent direction $\boldsymbol{d}_{des}$ should optimize or not damage each objective. Thus, moving against $\boldsymbol{d}_{des}$, starting from $\boldsymbol{\theta}$, amounts to a decrease in the objective values. When $\boldsymbol{d}_{des}^T \boldsymbol{g}_i = 0$, the optimization process ends, implying that no direction can further optimize the all the objectives simultaneously. At this moment, if we still want to improve the performance of a specific objective, other objectives will deteriorate.

This paper focuses on dynamically searching for a desired Pareto optimal solution with the aid of the proposed TCM loss in DA problems, aiming to enhance the performance on the target domain.

## 3 Related Work

**Domain Adaptation** aims to transfer knowledge across domains. The theory proposed by Ben-David et al. [2] reveals that the key ideology of DA is to diminish the domain gap between two domains, enabling successful transfer. Existing DA methods mainly mitigate domain shift via minimizing

some statistic metrics [26, 56, 28, 44] or adversarial training [11, 27, 41, 20, 22]. To name a few, DAN [26] mitigates the layer-wise discrepancy via Maximum Mean Discrepancy (MMD) metric. MDD [56] introduces Margin Disparity Discrepancy with rigorous generalization bounds to measure the discrepancy between domains. On the other hand, inspired by GANs [14], DANN [11] introduces a domain discriminator to enforce the features from two domains to be indistinguishable. CDAN [27] further extends this idea by integrate classification information into domain discriminator, leading to precise domain alignment. By contrast, we adopt domain discriminators to refine the target predictions with domain labels. In addition, these methods directly optimize the linear combination of all objectives, while overlooking on the potential conflict between them. In such case, the linear scheme is hard to tune and may only access to restricted solutions on the convex part of pareto front [4]. We propose a gradient-based optimization scheme to search the desirable Pareto optimal solution on the entire pareto front without weight hyper-parameters, which is orthogonal to prior DA methods.

**Weight-Based Optimization** is an intuitive strategy to optimize multiple objectives at the same time, which unifies all the training objectives using weight hyper-parameters to form an overall objective. Since it is hard to manually assign proper weights in prior, many DA methods conduct Deep Embedded Validation [53] to select weight hyper-parameters. Recently, some methods propose to dynamically weight the objectives leveraging gradient magnitude [6], the rate of change in losses [25], task uncertainty [19], or learning non-linear loss combinations by implicit differentiation [31]. However, these weight-based methods focus on seeking a balanced solution, which is not bound to be optimal for target classification in DA. By contrast, we explore the gradient-based optimization, aiming to find an optimal solution for DA problems.

**Multi-Objective Gradient-Based Optimization** leverages the gradients of objectives to reach the final Pareto optimal solution. MGDA [1] proposes to simply update the shared network parameters along a descent direction which leads to solutions that dominate the previous one. Nonetheless, MGDA may reach any Pareto optimal solution since it has no extra control for the optimization direction. To ameliorate this problem, PMTL [23] splits the loss space into separated sub-regions based on several preference vectors, and achieve Pareto optimal solutions in each sub-region of the Pareto front. Further, EPO [30] can reach a exact Pareto optimal solution by enforcing the decent direction correspond a specific preference vector. However, prior knowledge of which Pareto solution is the optimal one for target classification is often unavailable in DA. In this paper, we propose ParetoDA algorithm to dynamically searching the desirable Pareto optimal solution for target domain guided by a introduced TCM loss on a held-out target data .

## 4 Pareto Domain Adaptation

### 4.1 Decompose DA

Domain adaptation (DA) aims to train a model that performs well on an unlabeled target domain with the assistance of a labeled source domain. Formally, in DA, we can access a source domain $D_s = \{\boldsymbol{x}_i^s, y_i^s\}_{i=1}^{N_s}$ with $N_s$ labeled instances and a target domain $D_t = \{\boldsymbol{x}_i^t\}_{i=1}^{N_t}$ with $N_t$ unlabeled samples, where $y_i^s \in \{1, 2, ..., K\}$ is the corresponding label of the source data $\boldsymbol{x}_i^s$. Both source and target domains share one identical label space $\mathcal{Y}_s$ that contains $K$ classes.

Existing DA methods are mainly devoted to establishing domain alignment by explicitly minimizing the domain discrepancy or adversarial training, which have obtained promising results. Discrepancy-based methods attempt to find a metric that can precisely depict the domain discrepancy, then adopt it to minimize the domain gap. While adversarial-based methods play a mini-max game between a feature extractor and a discriminator, where the feature extractor tries to fool the discriminator, leading to domain-invariant features. All in all, these approaches can be decomposed as follows:

$$\text{Discrepancy-based: } \min_{\boldsymbol{\theta}, \boldsymbol{\phi}_c} \mathcal{L}_S + \mathcal{L}_D \tag{1}$$

$$\text{Adversarial-based: } \min_{\boldsymbol{\theta}, \boldsymbol{\phi}_c} \mathcal{L}_S + \mathcal{L}_D, \ \max_{\boldsymbol{\phi}_d} \mathcal{L}_D, \tag{2}$$

where $\mathcal{L}_S$ is the source classification loss, $\mathcal{L}_D$ is the domain alignment loss, and $\boldsymbol{\theta}, \boldsymbol{\phi}_c, \boldsymbol{\phi}_d$ are the parameters of shared feature extractor, classifier and domain discriminator, respectively.

However, the optimization of $\mathcal{L}_D$ may conflict with that of $\mathcal{L}_S$ due to the domain shift, under which circumstances the de facto optimization scheme can only access restricted Pareto optimal solutions.

Moreover, our goal is to minimize the target classification loss $\mathcal{L}_T^*$, but neither $\mathcal{L}_S$ or $\mathcal{L}_D$ directly corresponds to this goal. Therefore, to search for a desirable Pareto optimal solution for this goal, we propose a gradient-based optimization method for DA. Concretely, we first design a novel surrogate loss that mimics target classification loss, and then leverage the gradient of it on a held-out target dataset to guide the cooperative optimization of objectives.

## 4.2 Gradient-Based Optimization for DA

### 4.2.1 Target-Prediction Refining Mechanism

The goal of ParetoDA is to cooperatively optimize the multiple objectives in DA, aiming to minimize the target classification loss $\mathcal{L}_T^*$, which is inaccessible due to the absence of target labels. To tackle this issue, we design a TCM loss $\mathcal{L}_T$ by the mutual information [17] leveraging the target predictions. In this loss, target predictions act as (soft) labels for themselves. Therefore, the accuracy of the predictions is important to support the mimicking.

Former methods generally adopt the outputs of classifier as target predictions, without considering that the domain labels might be beneficial for classification. To refine the predictions by maximally exploiting the information at hand, we propose to model the domain label as conditional information into the prediction probability by Bayes' theorem. Formally, the class-conditional probability of sample $\boldsymbol{x}$ is $p(y = k \mid z = d, \boldsymbol{x}, \boldsymbol{\theta}, \boldsymbol{\phi}_c)$ instead of $p(y = k \mid \boldsymbol{x}, \boldsymbol{\theta}, \boldsymbol{\phi}_c)$, where the class $k \in \{1, \dots, K\}$ and the domain label $d \in \{0, 1\}$, $K$ is the number of classes, 0 and 1 denote the source and target domain respectively. Based on Bayes' theorem, we have:

$$
\begin{aligned}
p(y = k \mid z = d, \boldsymbol{x}, \boldsymbol{\theta}, \boldsymbol{\phi}_c) &= \frac{p(z = d \mid y = k, \boldsymbol{x}, \boldsymbol{\theta}, \boldsymbol{\phi}_c) p(y = k \mid \boldsymbol{x}, \boldsymbol{\theta}, \boldsymbol{\phi}_c)}{\sum_{k'} p(z = d \mid y = k', \boldsymbol{x}, \boldsymbol{\theta}, \boldsymbol{\phi}_c) p(y = k' \mid \boldsymbol{x}, \boldsymbol{\theta}, \boldsymbol{\phi}_c)} \\
&= \frac{p(z = d \mid \boldsymbol{x}, \boldsymbol{v}_k) p(y = k \mid \boldsymbol{x}, \boldsymbol{\theta}, \boldsymbol{\phi}_c)}{\sum_{k'} p(z = d \mid \boldsymbol{x}, \boldsymbol{v}_{k'}) p(y = k' \mid \boldsymbol{x}, \boldsymbol{\theta}, \boldsymbol{\phi}_c)} = \rho_{k|d}.
\end{aligned} \tag{3}
$$

Here, we adopt a multi-layer perception (MLP) network to model $p(z = d \mid \boldsymbol{x}, \boldsymbol{v}_k)$, called class-wise discriminator, where $\boldsymbol{v}_k$ denotes the parameters of the $k$-th domain discriminator corresponding to class $k$, which aims to distinguish the domain label of the samples from class $k$. $\rho_{k|d}$ can be regarded as a multiplicative ensemble of two predictions: $\frac{p(z=d|\boldsymbol{x},\boldsymbol{v}_k)}{\sum_{k'} p(z=d|\boldsymbol{x},\boldsymbol{v}_{k'})}$ and $\frac{p(y=k|\boldsymbol{x},\boldsymbol{\theta},\boldsymbol{\phi}_c)}{\sum_{k'} p(y=k'|\boldsymbol{x},\boldsymbol{\theta},\boldsymbol{\phi}_c)}$. We interpret how the former prediction works for target classification: for one specific class-wise discriminator, it can distinguish the target domain from the source well only when the input samples contain the corresponding class information, i.e., belong to the corresponding category. Otherwise, the probability of predicting samples as the target will be low.

We train all the class-wise discriminators with (derivations are in the supplementary material):

$$
\min_{\boldsymbol{v}_1, \dots, \boldsymbol{v}_K} - \sum_{k=1}^{K} \sum_{d=0}^{1} s_{k|d} I(z = d) \log p(z = d \mid \boldsymbol{x}, \boldsymbol{v}_k)|. \tag{4}
$$

Note that the $s_{k|d}$ here only acts as a weighting scalar, and for source domain $s_{k|d} = I(y = k)$ is hard class label (ground truth label) while for target domain, $s_{k|d} = \rho_{k|d}$ is soft class label (refined prediction). This objective is from an Expectation-Maximization derivation.

Now we acquire refined predictions $\rho_{k|d}$ for target samples which can substitute for the original predictions. As discussed in [12, 42], the ideal target predictions should be individually certain and globally diverse, which can be obtained by maximizing the mutual information between input samples and the predictions. Thus, we adopt the information maximization (IM) loss [17] on our improved version prediction $\rho_{k|d}$, the formulation is as follows:

$$
\mathcal{L}_T = \sum_{k=1}^{K} \hat{\rho}_{k|1} log \hat{\rho}_{k|1} - \mathbb{E}_{\boldsymbol{x} \in D_t} \sum_{k=1}^{K} \rho_{k|1} log \rho_{k|1}, \tag{5}
$$

where $\rho_{k|1}$ denotes $p(y = k \mid z = 1, \boldsymbol{x}, \boldsymbol{\theta})$, i.e., the probability that the target sample $\boldsymbol{x}$ is classified into class $k$ under the prior condition that it is divided into target domain, and $\hat{\rho}_{k|1}$ denotes $\mathbb{E}_{\boldsymbol{x} \in D_t} \rho_{k|1}$, represents the mean of the $k$-th elements in the outputs of the whole target domain.

Hence, we take $\mathcal{L}_T$ as the surrogate to mimic the target classification. This loss can be directly added to original DA methods to assist the model adaption to the target domain. More importantly, we leverage it to guide the search for the desirable Pareto optimal solution.

### 4.2.2 Dynamic Preference Mechanism

This section introduces how we establish gradient-based optimization for DA to handle the inconsistency among objectives and how we dynamically guide the optimization direction towards the desired Pareto optimal solution.

Given losses $\mathcal{L}_S$, $\mathcal{L}_D$ and $\mathcal{L}_T$, in each optimization step, we first model the update direction $\boldsymbol{d}$ as a convex combination of gradients of the three losses, i.e., $\boldsymbol{d} = \boldsymbol{G}\boldsymbol{w}$, where $\boldsymbol{w} \in \mathcal{S}^m = \{\boldsymbol{w} \in \mathbb{R}_+^m | \sum_{j=1}^m w_j = 1\}$, $m = 3$, and $\boldsymbol{G} = [\nabla_{\boldsymbol{\theta}}\mathcal{L}_S, \nabla_{\boldsymbol{\theta}}\mathcal{L}_D, \nabla_{\boldsymbol{\theta}}\mathcal{L}_T]$, where $\boldsymbol{\theta}$ are the shared parameters, i.e., the parameters of the shared feature extractor. A fundamental goal of gradient-based optimization is to find a $\boldsymbol{d}$ to simultaneously minimize all the losses. That is, the network can be optimized along a consistent direction and learn the shared knowledge between source and target domains efficiently and stably. Therefore, the direction $\boldsymbol{d}$ first need to satisfy the constraint $\boldsymbol{d}^T\boldsymbol{g}_j \geq 0, \forall j \in [m]$, where $\boldsymbol{g}_j$ is the $j$-th column of $G$.

Recently, a state-of-art method EPO [30] further controls the direction to reach a Pareto optimal solution with designated trade-off among objectives, using a preference vector fixed apriori. Concretely, EPO guides the optimization direction via minimizing the KL divergence between a normalized loss vector weighted by the preference vector and a uniform probability vector. Since the prior knowledge for the trade-off among objectives in DA is often unavailable and may vary with methods and training procedures, to weight $\mathcal{L}_T$, $\mathcal{L}_S$ and $\mathcal{L}_D$ to guide the cooperative optimization towards minimizing $\mathcal{L}_T^*$, we propose to model the gradient of $\mathcal{L}_T$ on a held-out unlabeled target dataset as the optimization guidance. We evaluate the performance on the held-out data because 1) it suggests generalization performance, and 2) it is more stable and convincible than training data. In practice, we randomly split 10% data from the original target set as the validation set and the rest 90% are taken as the training set. Note that we do not use any ground truth labels on the validation set, instead we calculate the proposed TCM loss on it, and then leverage this validation loss to guide the optimization direction.

Formally, we denote by $\mathcal{L}_{Val}$ the TCM loss on the held-out data, then we can directly obtain the gradient descent direction: $\hat{\boldsymbol{g}}_v = \nabla_{\boldsymbol{\theta}}\mathcal{L}_{Val}$. Thus we can replace the guidance direction $\boldsymbol{d}_{bal}$ [1] in EPO by $\hat{\boldsymbol{g}}_v$ as a dynamical guidance of the optimization direction. For clarity, the optimization problem can be formulated as a linear programming (LP) problem:

$$
\begin{aligned}
\boldsymbol{w}^* = \arg \max_{\boldsymbol{w} \in \mathcal{S}^m} \ & (\boldsymbol{G}\boldsymbol{w})^T (I(\mathcal{L}_{Val} > 0)\hat{\boldsymbol{g}}_v + I(\mathcal{L}_{Val} = 0)G\boldsymbol{1}/m), \\
\text{s.t. } & (\boldsymbol{G}\boldsymbol{w})^T \boldsymbol{g}_j \geq I(J \neq \emptyset)(\hat{\boldsymbol{g}}_v^T \boldsymbol{g}_j), \ \forall j \in \bar{J} - J^*, \\
& (\boldsymbol{G}\boldsymbol{w})^T \boldsymbol{g}_j \geq 0, \ \forall j \in J^*,
\end{aligned}
\tag{6}
$$

where $\mathcal{S}^m = \{\boldsymbol{w} \in \mathbb{R}_+^m | \sum_{j=1}^m w_j = 1\}$, $I(\cdot)$ is an indicator function, $\boldsymbol{1} \in \mathbb{R}^m$ is a vector whose elements are all 1, $J = \{j|\hat{\boldsymbol{g}}_v^T \boldsymbol{g}_j > 0\}$, $\bar{J} = \{j|\hat{\boldsymbol{g}}_v^T \boldsymbol{g}_j \leq 0\}$, and $J^* = \{j|\hat{\boldsymbol{g}}_v^T \boldsymbol{g}_j = \max_{j'} \hat{\boldsymbol{g}}_v^T \boldsymbol{g}_{j'}\}$. One may be concerned that involving the gradient of the held-out data in training may eventually cause over-fitting of the data. Fortunately, we use the following result to show that the held-out data can guide but will not be over-fitted by the optimization of our method.

**Theorem 1.** *Let $\boldsymbol{w}^*$ be the solution of the problem in Eq. (6), and $\boldsymbol{d}^* = \boldsymbol{G}\boldsymbol{w}^*$ be the resulted update direction. If $\mathcal{L}_{Val} = 0$, then the dominating direction $\boldsymbol{d}^*$ becomes a descent direction, i.e.,*

$$
(\boldsymbol{d}^*)^T \boldsymbol{g}_j \geq 0, \ \forall j \in \{1, 2, 3\}. \tag{7}
$$

*On the other hand, if $L_{Valid} > 0$, Let $\boldsymbol{\gamma}^* = (\boldsymbol{d}^*)^T \hat{\boldsymbol{g}}_v$ be the objective value of the problem in Eq. (6). Then,*

$$
\begin{aligned}
& if \ \boldsymbol{\gamma}^* > 0, \ then \ (\boldsymbol{d}^*)^T \hat{\boldsymbol{g}}_v > 0; \\
& if \ \boldsymbol{\gamma}^* \leq 0, \ then \ (\boldsymbol{d}^*)^T \boldsymbol{g}_j \geq 0, \ \forall j \in \{1, 2, 3\}.
\end{aligned}
\tag{8}
$$

The proof of Theorem 1 is deferred to the supplementary material. Note that we add a small $\epsilon = 1e-3$ to relax the condition $\mathcal{L}_{Val} > 0$ and $\mathcal{L}_{Val} = 0$ in Eq. (6) as $\mathcal{L}_{Val} > \epsilon$ and $\mathcal{L}_{Val} \leq \epsilon$ in practice, since the guiding significance of $\mathcal{L}_{Val}$ becomes trivial when it is less than a small value.

---

[1] See [30] for details.

Now we interpret the learning mechanism in Eq. (6). When $\mathcal{L}_{Val} = 0$, the learning is in the **pure descent mode**, in which the update direction $\boldsymbol{d}^* = \boldsymbol{G}\boldsymbol{w}^*$ approximates the mean gradient $\boldsymbol{G}\boldsymbol{1}/m$, and $\boldsymbol{d}^*$ can decrease all the training losses simultaneously. Whereas when $\mathcal{L}_{Val} > 0$, the learning is in the **guidance descent mode**, in which $\boldsymbol{d}^*$ approximates $\hat{\boldsymbol{g}}_v$. Since $\boldsymbol{d}^* = \boldsymbol{G}\boldsymbol{w}^*$ is a convex combination of columns in $G$, then $\gamma^* > 0$ means that $\hat{\boldsymbol{g}}_v$ is consistent with some columns in $G$, and $\boldsymbol{d}^*$ is forced to decrease the loss whose gradient is the most consistent with $\hat{\boldsymbol{g}}_v$—which means the highest contribu-

---

**Algorithm 1** Update Equations for ParetoDA

1: **Input:** Shared parameters $\boldsymbol{\theta}^t \in \mathbb{R}^d$, step size $\eta$
2: Calculate objective values: $\mathcal{L}_S, \mathcal{L}_D, \mathcal{L}_T$
3: Calculate guidance loss on target validation set: $\mathcal{L}_{val}$ (using Eq. (5))
4: Calculate the gradients of training objectives: $G = [g_1, g_2, g_3] = [\nabla_{\boldsymbol{\theta}^t} \mathcal{L}_S, \nabla_{\boldsymbol{\theta}^t} \mathcal{L}_D, \nabla_{\boldsymbol{\theta}^t} \mathcal{L}_T]$
5: Calculate the gradient of the guidance loss: $\hat{\boldsymbol{g}}_v = \nabla_{\boldsymbol{\theta}^t} \mathcal{L}_{val}$
6: Obtain $\boldsymbol{w}^*$ by solving LP (Eq. (6))
7: Optimization direction: $\boldsymbol{d}^* = \boldsymbol{G}\boldsymbol{w}^*$
8: **Output:** Updated parameters $\boldsymbol{\theta}^{t+1} = \boldsymbol{\theta}^t - \eta \boldsymbol{d}^*$

---

tion to the model performance. On the other hand, $\gamma^* \leq 0$ means none of the columns in $G$ is consistent with $\hat{\boldsymbol{g}}_v$. In this case, $\boldsymbol{d}^*$ is not forced to consistent with $\hat{\boldsymbol{g}}_v$, but only required to decrease the training losses, while approximating $\hat{\boldsymbol{g}}_v$ to the maximum feasible extent.

In summary, $\hat{\boldsymbol{g}}_v$ can dynamically guide the optimization direction towards the desired Pareto solution, but will not over-fit the held-out data, and we no longer need prior knowledge to set a preference vector. The theoretical guarantee for the existence of a Pareto optimal is deferred to the supplementary material. And the updating algorithm of every training iteration is presented in Alg. 1. Note that this optimize scheme is only applied to the shared parameters.

**Time Complexity**. Let $m \in \mathbb{Z}_+$ be the number of objective, and $d \in \mathbb{Z}_+$ the dimension of model parameters. The computation of $(\boldsymbol{G}\boldsymbol{w})^T \hat{\boldsymbol{g}}_v$ of $(\boldsymbol{G}\boldsymbol{w})^T \boldsymbol{G}$ both has run time $O(m^2 d)$. With the best LP solver [7], our LP (Eq. 6), that has $m$ variables and at most $2m + 1$ constraints, has a runtime [2] of $O^*(m^{2.38})$. As in DA, we usually have $d \gg m$, our total run time is $O(m^2 d + m^{2.38}) = O(d)$.

**Theoretical Insight**. In Domain adaptation literature, the theory proposed by Ben-David et al. [2] provides an upper bound for the generalization error on target data, which is composed of three terms: 1) the expected error on source domain $\epsilon_S(h)$; 2) the $H \Delta H$ divergence between two domains $d_{H\Delta H}(D_s, D_t)$; 3) the combined error $\lambda$ of the ideal joint hypothesis $h^*$. The formulation is:

$$\epsilon_T(h) \leq \epsilon_S(h) + \frac{1}{2} d_{H\Delta H}(D_s, D_t) + \lambda, \tag{9}$$

where $\lambda = \epsilon_S(h^*) + \epsilon_T(h^*)$ and $h^* = \arg\min_{h \in H} \epsilon_S(h) + \epsilon_T(h)$.

The goal of DA methods is to lower the upper bound of the expected target error. $\epsilon_S(h)$ is expected to be small thanks to the supervision in source domain. And $\lambda$ is also generally considered sufficiently small [41, 27]. Thus, most DA methods mainly focus on reducing $d_{H\Delta H}(D_s, D_t)$ by domain alignment. However, if one of the three terms goes down and another term goes up, the model still can not effectively reduce the upper bound of $\epsilon_T(h)$. Excessive either source classification or domain alignment will cause damage to the other.

In our ParetoDA, by adopting gradient-based optimization, we can simultaneously optimize the source classification loss and the domain alignment loss, so as to make $\epsilon_S(h)$ and $d_{H\Delta H}(D_s, D_t)$ decrease synchronously. Moreover, both the training of our proposed TCM loss on target data and the optimization guidance of this loss on held-out data contribute to the classification ability of the model to the target domain, and then reduce the combined error $\lambda$ of the ideal joint hypothesis $h^*$. Consequently, the upper bound of $\epsilon_T(h)$ can be effectively reduced in our work, which validates the effectiveness of ParetoDA theoretically.

# 5 Experiment

## 5.1 Dataset and Setup

We evaluate ParetoDA on three object classification datasets and two semantic segmentation datasets. **Office-31 [40]** is a typical benchmark for cross-domain object classification involving three distinct

---

[2]$O^*$ is used to hide $m^{o(1)}$ and $log^{O(1)}(1/\delta)$ factors, $\delta$ being the relative accuracy. See [7] for details.

Table 2: Accuracy(%) on **Office-Home** for unsupervised DA (ResNet-50).

| Method | Ar:Cl | Ar:Pr | Ar:Rw | Cl:Ar | Cl:Pr | Cl:Rw | Pr:Ar | Pr:Cl | Pr:Rw | Rw:Ar | Rw:Cl | Rw:Pr | Avg. |
|---|---|---|---|---|---|---|---|---|---|---|---|---|---|
| ResNet [15] | 34.9 | 50.0 | 58.0 | 37.4 | 41.9 | 46.2 | 38.5 | 31.2 | 60.4 | 53.9 | 41.2 | 59.9 | 46.1 |
| DAN [26] | 43.6 | 57.0 | 67.9 | 45.8 | 56.5 | 60.4 | 44.0 | 43.6 | 67.7 | 63.1 | 51.5 | 74.3 | 56.3 |
| TAT [24] | 51.6 | 69.5 | 75.4 | 59.4 | 69.5 | 68.6 | 59.5 | 50.5 | 76.8 | 70.9 | 56.6 | 81.6 | 65.8 |
| TPN [35] | 51.2 | 71.2 | 76.0 | 65.1 | 72.9 | 72.8 | 55.4 | 48.9 | 76.5 | 70.9 | 53.4 | 80.4 | 66.2 |
| BNM [9] | 52.3 | 73.9 | 80.0 | 63.3 | 72.9 | **74.9** | 61.7 | 49.5 | 79.7 | 70.5 | 53.6 | 82.2 | 67.9 |
| MDD [56] | 54.9 | 73.7 | 77.8 | 60.0 | 71.4 | 71.8 | 61.2 | 53.6 | 78.1 | 72.5 | 60.2 | 82.3 | 68.1 |
| GSP [51] | **56.8** | 75.5 | 78.9 | 61.3 | 69.4 | **74.9** | 61.3 | 52.6 | 79.9 | 73.3 | 54.2 | 83.2 | 68.4 |
| DANN [11] | 45.6 | 59.3 | 70.1 | 47.0 | 58.5 | 60.9 | 46.1 | 43.7 | 68.5 | 63.2 | 51.8 | 76.8 | 57.6 |
| +BSP [5] | 51.4 | 68.3 | 75.9 | 56.0 | 67.8 | 68.8 | 57.0 | 49.6 | 75.8 | 70.4 | 57.1 | 80.6 | 64.9 |
| +MetaAlign [50] | 48.6 | 69.5 | 76.0 | 58.1 | 65.7 | 68.3 | 54.9 | 44.4 | 75.3 | 68.5 | 50.8 | 80.1 | 63.3 |
| +ParetoDA | 55.2 | 74.4 | 79.0 | 61.9 | 72.4 | 72.9 | 62.1 | **55.8** | 81.1 | 74.4 | **61.1** | 82.4 | 69.4 |
| CDAN [27] | 50.7 | 70.6 | 76.0 | 57.6 | 70.0 | 70.0 | 57.4 | 50.9 | 77.3 | 70.9 | 56.7 | 81.6 | 65.8 |
| +BSP [5] | 52.0 | 68.6 | 76.1 | 58.0 | 70.3 | 70.2 | 58.6 | 50.2 | 77.6 | 72.2 | 59.3 | 81.9 | 66.3 |
| +MetaAlign [50] | 55.2 | 70.5 | 77.6 | 61.5 | 70.0 | 70.0 | 58.7 | 55.7 | 78.5 | 73.3 | 61.0 | 81.7 | 67.8 |
| +ParetoDA | **56.8** | **75.9** | **80.5** | **64.4** | **73.5** | 73.7 | **65.6** | 55.2 | **81.3** | **75.2** | **61.1** | **83.9** | **70.6** |

domain: Amazon (A), DSLR (D) and Webcam (W); **Office-Home** [48] is a more difficult benchmark composed by four image domains: Art (Ar), Clip Art (Cl), Product (Pr) and Real-World (Rw). The images of different domains are more visually dissimilar with each other; **VisDA-2017** [37] is a large-scale synthetic-to-real dataset, containing over 280K images across 12 categories; **Cityscapes** [8] is a real-world semantic segmentation dataset with 5,000 urban scenes which are divided into training, validation and test splits; **GTA5** [38] includes 24,966 game screenshots from the game engine of GTA5. Following [46, 20], we establish a transfer semantic segmentation scenario: GTA5 → Cityscapes. Specifically, we use the shared 19 categories and report the test results on the validation set of Cityscapes. To evaluate our approach, we apply the proposed ParetoDA on former DA methods, i.e., DANN, CDAN, and MDD, and split 10% data randomly from the original target set as the validation set. All experiments are implemented via PyTorch [36], and we adopt the common-used stochastic gradient descent (SGD) optimizer with momentum 0.9 and weight decay 1e-4 for all experiments. Following the standard protocols in [27, 11, 29], we take the average classification accuracy and mean IoU (mIoU) for image classification and semantic segmentation as evaluation metrics, respectively. More details are presented in the supplementary material due to limited space.

## 5.2 Experimental Results

**Office-31**. From the results in Table 1, we can observe that DA methods, such as DANN and CDAN exceed the ResNet model. This validates that mitigating the domain gap between two domains is crucial for enhancing the generalization performance of models. We evaluate ParetoDA on three basic DA methods. The results show that ParetoDA can consistently improve the performance of the basic methods significantly. In particular, ParetoDA improves DANN by 8.0%, revealing that ParetoDA is an extra booster to former DA methods. On the other hand, the encouraging results validate the importance of searching for a desirable Pareto optimal solution.

Table 1: Classification accuracy (%) on Office-31 (ResNet-50).

| Method | A:W | D:W | W:D | A:D | D:W | W:A | Avg |
|---|---|---|---|---|---|---|---|
| ResNet [15] | 68.4 | 96.7 | 99.3 | 68.9 | 62.5 | 60.7 | 76.1 |
| DAN [26] | 80.5 | 97.1 | 99.6 | 78.6 | 63.6 | 62.8 | 80.4 |
| JADA [22] | 90.5 | 97.5 | 100.0 | 88.2 | 70.9 | 70.6 | 86.1 |
| BNM [9] | 91.5 | 98.5 | 100.0 | 90.3 | 70.9 | 71.6 | 87.1 |
| TAT [24] | 92.5 | 99.3 | 100.0 | 93.2 | 73.1 | 72.1 | 88.4 |
| GSP [51] | 92.9 | 98.7 | 99.8 | 94.5 | 75.9 | 74.9 | 89.5 |
| DANN [11] | 82.0 | 96.9 | 99.1 | 79.7 | 68.2 | 67.4 | 82.2 |
| +BSP [5] | 93.0 | 98.0 | 100.0 | 90.0 | 71.9 | 73.0 | 87.7 |
| +MCC [18] | 95.6 | 98.6 | 99.3 | 93.8 | 74.0 | 75.0 | 89.4 |
| +ParetoDA | **95.5** | 98.7 | 100.0 | 93.8 | 76.7 | 76.3 | 90.2 |
| CDAN [27] | 94.1 | 98.6 | 100.0 | 92.9 | 71.0 | 69.3 | 87.7 |
| +BSP [5] | 93.3 | 98.2 | 100.0 | 93.0 | 73.6 | 72.6 | 88.5 |
| +MCC [18] | 94.7 | 98.6 | 100.0 | 95.0 | 73.0 | 73.6 | 89.2 |
| +ParetoDA | 95.0 | **98.9** | 100.0 | 95.4 | **77.6** | 75.7 | **90.4** |
| MDD [56] | 93.5 | 98.4 | 100.0 | 94.5 | 74.6 | 72.2 | 88.9 |
| +ParetoDA | 95.4 | **98.9** | 100.0 | 94.4 | 76.2 | **75.8** | 90.1 |

**Office-Home**. Compared with Office-31, Office-Home is a more difficult dataset, whose results are shown in Table 2. In this difficult scenario, ParetoDA can still enhance the performance of former DA methods, with 11.8% and 4.8% improvement on DANN and CDAN respectively. Based on these promising results, we can infer that ParetoDA can reach a desirable Pareto optimal solution which is beneficial for the performance of models.

Table 3: Accuracy(%) on **VisDA-2017** for unsupervised DA (ResNet-101).

| Method | plane | bcycl. | bus | car | horse | knife | mcycl. | person | plant | sktbrd | train | truck | Avg. |
|---|---|---|---|---|---|---|---|---|---|---|---|---|---|
| ResNet [15] | 55.1 | 53.3 | 61.9 | 59.1 | 80.6 | 17.9 | 79.7 | 31.2 | 81.0 | 26.5 | 73.5 | 8.5 | 52.4 |
| DAN [26] | 87.1 | 63.0 | 76.5 | 42.0 | 90.3 | 42.9 | 85.9 | 53.1 | 49.7 | 36.3 | 85.8 | 20.7 | 61.1 |
| MCD [41] | 87.0 | 60.9 | 83.7 | 64.0 | 88.9 | 79.6 | 84.7 | 76.9 | 88.6 | 40.3 | 83.0 | 25.8 | 71.9 |
| JADA [22] | 91.9 | 78.0 | 81.5 | 68.7 | 90.2 | 84.1 | 84.0 | 73.6 | 88.2 | 67.2 | 79.0 | 38.0 | 77.0 |
| TPN [35] | 93.7 | **85.1** | 69.2 | 81.6 | 93.5 | 61.9 | 89.3 | 81.4 | 93.5 | 81.6 | 84.5 | 49.9 | 80.4 |
| DTA [21] | 93.7 | 82.2 | **85.6** | **83.8** | 93.0 | 81.0 | **90.7** | 82.1 | **95.1** | 78.1 | 86.4 | 32.1 | 81.5 |
| DANN [11] | 81.9 | 77.7 | 82.8 | 44.3 | 81.2 | 29.5 | 65.1 | 28.6 | 51.9 | 54.6 | 82.8 | 7.8 | 57.4 |
| +BSP [5] | 92.2 | 72.5 | 83.8 | 47.5 | 87.0 | 54.0 | 86.8 | 72.4 | 80.6 | 66.9 | 84.5 | 37.1 | 72.1 |
| +ParetoDA | 95.3 | 82.4 | 82.9 | 56.2 | 92.6 | **95.0** | 87.1 | 81.1 | 90.2 | 90.8 | **87.8** | 46.7 | 82.4 |
| CDAN [27] | 85.2 | 66.9 | 83.0 | 50.8 | 84.2 | 74.9 | 88.1 | 74.5 | 83.4 | 76.0 | 81.9 | 38.0 | 73.9 |
| +BSP [5] | 92.4 | 61.0 | 81.0 | 57.5 | 89.0 | 80.6 | 90.1 | 77.0 | 84.2 | 77.9 | 82.1 | 38.4 | 75.9 |
| +ParetoDA | **95.9** | 82.8 | 81.3 | 58.7 | **93.9** | 93.7 | 85.9 | **83.0** | 91.9 | **92.0** | 87.1 | **51.8** | **83.2** |

Table 4: Semantic segmentation performance mIoU (%) on **Cityscapes** validation set.

| Method | road | side. | buil. | wall | fence | pole | light | sign | veg | terr. | sky | pers. | rider | car | truck | bus | train | mbike | bike | mIoU |
|---|---|---|---|---|---|---|---|---|---|---|---|---|---|---|---|---|---|---|---|---|
| ResNet-101 | 75.8 | 16.8 | 77.2 | 12.5 | 21.0 | 25.5 | 30.1 | 20.1 | 81.3 | 24.6 | 70.3 | 53.8 | 26.4 | 49.9 | 17.2 | 25.9 | 6.5 | 25.3 | 36.0 | 36.6 |
| AdaSegNet [46] | 86.5 | 36.0 | 79.9 | 23.4 | 23.3 | 23.9 | 35.2 | 14.8 | 83.4 | 33.3 | 75.6 | 58.5 | 27.6 | 73.7 | 32.5 | 35.4 | 3.9 | 30.1 | 28.1 | 42.4 |
| Cycada [16] | 86.7 | 35.6 | 80.1 | 19.8 | 17.5 | 38.0 | 39.9 | 41.5 | 82.7 | 27.9 | 73.6 | **64.9** | 19 | 65.0 | 12.0 | 28.6 | 4.5 | 31.1 | 42.0 | 42.7 |
| CLAN [29] | 87.0 | 27.1 | 79.6 | 27.3 | 23.3 | 28.3 | 35.5 | 24.2 | 83.6 | 27.4 | 74.2 | 58.6 | 28.0 | 76.2 | 33.1 | 36.7 | 6.7 | 31.9 | 31.4 | 43.2 |
| AdvEnt [49] | 89.9 | 36.5 | 81.6 | 29.2 | 25.2 | 28.5 | 32.3 | 22.4 | 83.9 | **34.0** | 77.1 | 57.4 | 27.9 | 83.7 | 29.4 | 39.1 | 1.5 | 28.4 | 23.3 | 43.8 |
| +ParetoDA | 79.6 | 29.1 | 83.4 | 30.4 | 22.4 | 32.6 | 39.2 | 26.7 | 82.5 | 26.5 | **82.2** | 63.0 | 29.8 | **85.9** | 36.1 | 43.5 | 9.2 | 29.9 | 43.7 | 46.1 |
| FDA [52] | **90.0** | 40.5 | 79.4 | 25.3 | 26.7 | 30.6 | 31.9 | 29.3 | 79.4 | 28.8 | 76.5 | 56.4 | 27.5 | 81.7 | 27.7 | 45.1 | 17.0 | 23.8 | 29.6 | 44.6 |
| +ParetoDA | 85.4 | 37.6 | **84.0** | 32.0 | 23.8 | 33.1 | 39.6 | 25.9 | **84.0** | 30.6 | 81.6 | 61.3 | 30.9 | 85.4 | 31.9 | 45.0 | 7.6 | **32.4** | 45.4 | 47.2 |
| CBST [59] | 86.8 | 46.7 | 76.9 | 26.3 | 24.8 | 42.0 | 46.0 | 38.6 | 80.7 | 15.7 | 48.0 | 57.3 | 27.9 | 78.2 | 24.5 | **49.6** | **17.7** | 25.5 | 45.1 | 45.2 |
| +ParetoDA | 82.4 | **50.7** | 80.2 | **34.9** | **36.7** | **45.9** | **47.2** | **45.1** | 72.9 | 27.8 | 46.0 | 62.3 | **33.0** | 81.4 | 29.5 | 44.3 | 14.0 | 29.8 | **49.3** | **48.1** |

**VisDA-2017**. To validate the model performance on a large-scale dataset, we carry out experiments on VisDA-2017. The results are presented in Table 3. As a plug-in module, BSP improves the basic models by enhancing the discriminability of feature representations. ParetoDA aims to improve the optimization for DA methods, which is also orthogonal to former DA methods. In addition, ParetoDA achieves greater performance gains, further indicating the effectiveness and versatility of our method.

**GTA5 $\rightarrow$ Cityscapes**. Compared with classification tasks, segmentation demands more cost for the pixel-level annotation. Hence, it's of vital importance to conduct domain adaptation for semantic segmentation. We evaluate our method on the adaptive semantic segmentation task, GTA5 $\rightarrow$ Cityscapes, aiming to transfer the knowledge from synthesized frames to real scenes. The segmentation results are exhibited in Table 4. We apply ParetoDA on two methods, i.e., AdvEnt and FDA, which boosts them by large margins on average. This manifests that ParetoDA can still reach an appropriate optimization solution for enhancing model performance on this pixel-level transfer scenario. Besides, ParetoDA is not limited to handle the conflict between the classical source classification loss and domain alignment loss, our framework can also be applied to where the explicit domain alignment module doesn't exist or there exist multiple losses. For instance, we adopt ParetoDA on a self-training based domain adaptation method CBST [59] and it is clear that ParetoDA is still effective.

## 5.3 Analytical Experiments

**Ablation Study.** To verify each component of ParetoDA, we establish ablation study on Office-31. The average results are shown in Table 5. TCM performs better than IM, demonstrating that integrating domain labels is beneficial for target predictions. Applying the conventional Pareto optimization method [30] with fixed preference vector on DANN achieves considerable performance gain. This reveals that the gradient-based optimization method is superior to the linear weighting scheme. Further, with our proposed

Table 5: Ablation Study of ParetoDA on Office-31 (ResNet-50).

| | IM | TCM | EPO | ParetoDA | Accuracy |
|---|---|---|---|---|---|
| DANN | | | | | 82.2 |
| DANN | ✓ | | | | 87.9 |
| DANN | | ✓ | | | 88.4 |
| DANN | | | ✓ | | 87.7 |
| DANN | ✓ | | | ✓ | 89.7 |
| DANN | | ✓ | ✓ | | 89.9 |
| DANN | | ✓ | | ✓ | **90.2** |

ParetoDA, DANN obtains the best results, validating the importance of dynamical preference learning and that TCM loss serves as a better guidance for optimization.

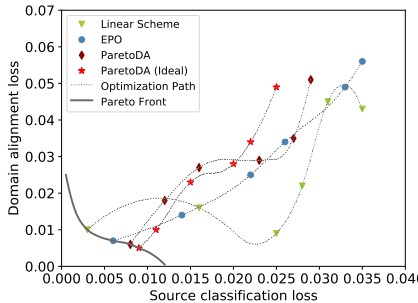
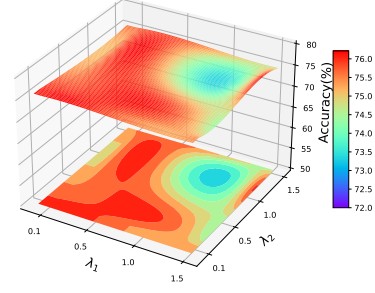

(a) Visualization of the optimization path     (b) Objective scale sensitivity analysis

Figure 3: Analytical experiments conducted on W→A (Office-31) based on DANN. (a): Visualization of optimization paths of different optimization schemes. (b): Objective scale sensitivity analysis of ParetoDA by varying the scale factors of objectives.

**Optimization Path Visualization.** To intuitively verify that ParetoDA can dynamically search the desired Pareto optimal solution under the guidance of our proposed loss, we visualize the optimization paths of different optimization schemes (see Fig. 3(a)). Note that all these schemes start from the same initial point, but we only plot several solutions of the final stage of optimization for better illustration. It is clear that the linear weighting scheme not only deteriorates one of the objectives at some steps, but also reaches a solution on the convex part of the Pareto front, which deviates severely from the ideal one. The conventional gradient-based optimization method EPO (with fixed preference vector $r = [1, 1, 1]$) controls the optimization direction to avoid damaging any of the objectives. However, its solution heavily relies on the pre-defined preference vector for training objectives, which is ambiguous in DA. By contrast, our proposed ParetoDA further dynamically adjusts the optimization direction with the gradient direction of the proposed TCM loss. For comparisons, we illustrate the **ideal** solution guided by the supervised classification loss on target domain. One can observe that the optimization direction of ParetoDA is consistent with the ideal one in general, and reach a final solution closest to the ideal one.

**Sensitivity to Different Objective Scales.** The scale of the objectives may vary with methods or datasets. To test the adaptivity of ParetoDA to different objective scales, we conduct experiments on a random task W→A (Office-31) based on DANN by varying $\lambda_0, \lambda_1 \in \{0.1, 0.5, 1.0, 1.5\}$, which represent the scale factor of object classification objective and domain alignment objective respectively. Fig. 3(b) shows that ParetoDA is not sensitive to the scale variety of objectives, which indicating that ParetoDA is robust across different methods and datasets.

**Adaptivity to Deeper Backbone Networks.** In this work, we reach a desirable Pareto optimal solution for the target domain by controlling the optimization direction. Whereas on deeper networks, it might be more difficult to control the optimization direction due to the increased complexity. For a fair comparison, we use ResNet-50 on classification tasks. However, ParetoDA can be easily adapted to deeper backbone networks. Table 6 presents the results of ParetoDA based on DANN with different backbones, demonstrating that ParetoDA can still consistently enhance the performance of the basic method.

Table 6: Test top-1 accuracy (%) on task W→A (Office-31) of methods with different backbone networks.

| Networks | Source-only | DANN [11] | +ParetoDA |
|---|---|---|---|
| ResNet-50 | 60.7 | 67.4 | 76.3 |
| ResNet-101 | 65.8 | 74.5 | 77.5 |
| ResNet-152 | 68.2 | 76.1 | 79.0 |

## 6 Conclusion

In this paper, we rethink the optimization scheme for DA from a gradient-based optimization perspective. We present a Pareto Domain Adaptation (ParetoDA) to dynamically search a desirable Pareto optimal solution for boosting model performance, which is orthogonal to former DA methods. Extensive experiments on both classification and segmentation tasks validate the effectiveness of ParetoDA. It is worth noting that we mainly concentrate on the traditional domain adaptation setting for clarity, while the generalization to other variants of DA needs to be further explored.

## Acknowledgments

This work was supported by the National Natural Science Foundation of China (61902028).

## Funding Transparency Statement

Funding in direct support of this work: National Natural Science Foundation of China (61902028).

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
