# Supplementary Material for: Pareto Domain Adaptation

**Fangrui Lv,**[1,*]   **Jian Liang,**[2,*]   **Kaixiong Gong,**[1]   **Shuang Li,**[1,†]

**Chi Harold Liu**,[1]   **Han Li**,[2]   **Di Liu**,[2]   **Guoren Wang**[1]

[1] Beijing Institute of Technology, China     [2]Alibaba Group, China

[1] {fangruilv,kxgong,shuangli,wanggrbit}@bit.edu.cn, liuchi02@gmail.com
[2] {xuelang.lj,lihan.lh,wendi.ld}@alibaba-inc.com

## 1   Potential Negative Societal Impacts

Our work focuses on domain adaptation and attempts to properly handle the multiple objectives optimization in it from a gradient-based perspective, which further enhance the performance of adaptive models. This method exerts a positive influence on the society and the community, saves the cost and time of data annotation, boosts the reusability of knowledge across domains, and greatly improves the efficiency. However, this work suffers from some negative consequences, which is worthy of further research and exploration. Specifically, more jobs of classification or target detection for rare or variable conditions may be cancelled. Besides, we should be cautious of the result of the failure of the system, which could render people believe that classification was unbiased. Still, it might be not, which might be misleading, e.g., when using the system in an unlabeled domain.

## 2   Experimental Details

We evaluate our method on cross-domain image classification and semantic segmentation datasets. For the former, we conduct experiments on Office-31, VisDA-2017 and Office-Home datasets. For the latter, we conduct experiments on GTA5 and Cityscapes. We directly quote the results from original papers if the experimental settings are the same. In addition, we report the average result of three random experiments.

### 2.1   Experiment on image classification

**Network architectures** We adopt the ResNet-50 and ResNet-101 [7] model pre-trained on ImageNet [9] as the backbone network for classification tasks, and replace the last fully connected layer with a bottleneck layer to speed up the experiments [6, 14]. The architecture of classifier is simply a single fully connected layers with random initialization attached to the bottleneck layer. And the domain discriminator is composed of three fully connected layers, each followed by a BatchNormalize layer and a Relu activation function. Admittedly, the choice for the classifier and discriminator is arbitrary and better adaptation performance may be obtained if this part of the architecture is tuned.

**Training details** We fine-tune from ImageNet pre-trained models and train other layers from the scratch with learning rate 10 times that of the feature generator layers. We adopt Stochastic Gradient Descent optimizer (SGD) [1] with learning rate 0.003, momentum 0.9 and weight decay $1 \times 10^{-4}$. Besides, we adopt the annealing procedure mentioned in [14] to schedule the learning rate. All experiments run on a single NVIDIA RTX 3090 GPU for training.

35th Conference on Neural Information Processing Systems (NeurIPS 2021).

Table 1: Classification accuracy (%) on Office-31 (ResNet-50).

| Method | A:W | D:W | W:D | A:D | D:W | W:A | Avg |
|---|---|---|---|---|---|---|---|
| ResNet [7] | 68.4 | 96.7 | 99.3 | 68.9 | 62.5 | 60.7 | 76.1 |
| DAN [13] | 80.5 | 97.1 | 99.6 | 78.6 | 63.6 | 62.8 | 80.4 |
| JADA [10] | 90.5 | 97.5 | **100.0** | 88.2 | 70.9 | 70.6 | 86.1 |
| BNM [4] | 91.5 | 98.5 | **100.0** | 90.3 | 70.9 | 71.6 | 87.1 |
| TAT [12] | 92.5 | 99.3 | **100.0** | 93.2 | 73.1 | 72.1 | 88.4 |
| GSP [19] | 92.9 | 98.7 | 99.8 | 94.5 | 75.9 | 74.9 | 89.5 |
| DANN [6] | 82.0 | 96.9 | 99.1 | 79.7 | 68.2 | 67.4 | 82.2 |
| +BSP [3] | 93.0 | 98.0 | **100.0** | 90.0 | 71.9 | 73.0 | 87.7 |
| +MCC [8] | 95.6 | 98.6 | 99.3 | 93.8 | 74.0 | 75.0 | 89.4 |
| +ParetoDA | **95.5 ± 0.1** | 98.7 ± 0.3 | **100.0 ± 0.0** | 93.8 ±0.2 | 76.7±0.2 | 76.3 ±0.3 | 90.2 |
| CDAN [14] | 94.1 | 98.6 | **100.0** | 92.9 | 71.0 | 69.3 | 87.7 |
| +BSP [3] | 93.3 | 98.2 | **100.0** | 93.0 | 73.6 | 72.6 | 88.5 |
| +MCC [8] | 94.7 | 98.6 | **100.0** | 95.0 | 73.0 | 73.6 | 89.2 |
| +ParetoDA | 95.0 ± 0.2 | **98.9±0.1** | **100.0 ±0.0** | **95.4 ± 0.2** | **77.6 ± 0.3** | 75.7 ± 0.2 | **90.4** |
| MDD [20] | 93.5 | 98.4 | **100.0** | 94.5 | 74.6 | 72.2 | 88.9 |
| +ParetoDA | 95.4 ± 0.2 | **98.9 ± 0.2** | **100.0 ± 0.0** | 94.4 ± 0.3 | 76.2 ± 0.4 | **75.8 ± 0.2** | 90.1 |

Table 2: Accuracy(%) on **Office-Home** for unsupervised DA (ResNet-50).

| Method | Ar:Cl | Ar:Pr | Ar:Rw | Cl:Ar | Cl:Pr | Cl:Rw | Pr:Ar | Pr:Cl | Pr:Rw | Rw:Ar | Rw:Cl | Rw:Pr | Avg. |
|---|---|---|---|---|---|---|---|---|---|---|---|---|---|
| ResNet [7] | 34.9 | 50.0 | 58.0 | 37.4 | 41.9 | 46.2 | 38.5 | 31.2 | 60.4 | 53.9 | 41.2 | 59.9 | 46.1 |
| DAN [13] | 43.6 | 57.0 | 67.9 | 45.8 | 56.5 | 60.4 | 44.0 | 43.6 | 67.7 | 63.1 | 51.5 | 74.3 | 56.3 |
| TAT [12] | 51.6 | 69.5 | 75.4 | 59.4 | 69.5 | 68.6 | 59.5 | 50.5 | 76.8 | 70.9 | 56.6 | 81.6 | 65.8 |
| TPN [16] | 51.2 | 71.2 | 76.0 | 65.1 | 72.9 | 72.8 | 55.4 | 48.9 | 76.5 | 70.9 | 53.4 | 80.4 | 66.2 |
| BNM [4] | 52.3 | 73.9 | 80.0 | 63.3 | 72.9 | **74.9** | 61.7 | 49.5 | 79.7 | 70.5 | 53.6 | 82.2 | 67.9 |
| MDD [20] | 54.9 | 73.7 | 77.8 | 60.0 | 71.4 | 71.8 | 61.2 | 53.6 | 78.1 | 72.5 | 60.2 | 82.3 | 68.1 |
| GSP [19] | **56.8** | 75.5 | 78.9 | 61.3 | 69.4 | **74.9** | 61.3 | 52.6 | 79.9 | 73.3 | 54.2 | 83.2 | 68.4 |
| DANN [6] | 45.6 | 59.3 | 70.1 | 47.0 | 58.5 | 60.9 | 46.1 | 43.7 | 68.5 | 63.2 | 51.8 | 76.8 | 57.6 |
| +BSP [3] | 51.4 | 68.3 | 75.9 | 56.0 | 67.8 | 68.8 | 57.0 | 49.6 | 75.8 | 70.4 | 57.1 | 80.6 | 64.9 |
| +MetaAlign [18] | 48.6 | 69.5 | 76.0 | 58.1 | 65.7 | 68.3 | 54.9 | 44.4 | 75.3 | 68.5 | 50.8 | 80.1 | 63.3 |
| +ParetoDA | 55.2 ± 0.4 | 74.4 ± 0.3 | 79.0 ± 0.4 | 61.9 ± 0.6 | 72.4 ± 0.5 | 72.9 ± 0.3 | 62.1 ± 0.4 | 55.8 ± 0.5 | 81.1 ± 0.4 | 74.4 ± 0.6 | 61.1 ± 0.5 | 82.4 ± 0.4 | 69.4 |
| CDAN [14] | 50.7 | 70.6 | 76.0 | 57.6 | 70.0 | 70.0 | 57.4 | 50.9 | 77.3 | 70.9 | 56.7 | 81.6 | 65.8 |
| +BSP [3] | 52.0 | 68.6 | 76.1 | 58.0 | 70.3 | 70.2 | 58.6 | 50.2 | 77.6 | 72.2 | 59.3 | 81.9 | 66.3 |
| +MetaAlign [18] | 55.2 | 70.5 | 77.6 | 61.5 | 70.0 | 70.0 | 58.7 | 55.7 | 78.5 | 73.3 | 61.0 | 81.7 | 67.8 |
| +ParetoDA | **56.8 ± 0.3** | **75.9 ± 0.4** | **80.5 ± 0.2** | **64.4 ± 0.5** | **73.5 ± 0.3** | 73.7 ± 0.6 | **65.6 ± 0.4** | 55.2 ± 0.5 | **81.3 ± 0.2** | **75.2 ± 0.5** | **61.1 ± 0.4** | **83.9 ± 0.3** | **70.6** |

## 2.2 Experiment on semantic segmentation

**Network architectures** We leverage the DeepLab-v2 [2] framework with the pre-trained ResNet-101 model as the base feature extractor for segmentation task. To better capture the scene context, Atrous Spatial Pyramid Pooling (ASPP) is used for classifier and applied on the $conv5$ feature outputs. We fix the sampling rates as {6, 12, 18, 24} following [17] and modify the stride and dilation rate of the last layers to produce denser feature maps with larger field-of-views.

**Training details** For evaluation metrics, we report pre-class intersection-over-union (IoU) and mean IoU over all classes. We use the evaluation code released along with the Cityscapes dataset, which calculates the PASCAL VOC intersection-over-union, i.e, $IoU = \frac{TP}{TP+FP+FN}$ [5], where $TP$, $FP$, $FN$ are the amount of true positive, false positive and false negative pixels, respectively, determined over the whole test set.

# 3   Experimental Results with Error Bars

For the sake of objective, we run all the experiments multiple times with random seed. We report the average results in the main body of paper for elegant, and show the complete results with error bars in the form of mean±std below (Table. 1,2).

# 4 Derivation and Proofs

## 4.1 Proof for Theorem 1

*Proof.* If $\mathcal{L}_{Val} = 0$, then its gradient is also a zero vector: $\hat{g}_v = \mathbf{0}$. This means $J$ is empty. As a result, $I(J \neq \emptyset) = 0$ and all the constraints of Eq. (6) becomes

$$(\boldsymbol{d}^*)^T \boldsymbol{g}_j \geq 0, \ \forall j \in \{1, 2, 3\}. \tag{1}$$

Then $\boldsymbol{d}^*$ becomes a descent direction.

On the other hand, we consider the case in that $\mathcal{L}_{Val} > 0$. Since $\gamma^* = (\boldsymbol{d}^*)^T \hat{g}_v$, then $\gamma^* > 0$ means $(\boldsymbol{d}^*)^T \hat{g}_v > 0$. Whereas if $\gamma^* = (\boldsymbol{d}^*)^T \hat{g}_v = (\boldsymbol{G}\boldsymbol{w}^*)^T \hat{g}_v \leq 0$, since $\boldsymbol{G}\boldsymbol{w}^*$ is the maximum of the convex combination of the columns in $\boldsymbol{G}$, it means that there is no gradient $\boldsymbol{g}_j$, $j \in \{1, 2, 3\}$, for which $\boldsymbol{g}_j^T \hat{g}_v > 0$. As a result, $J$ is empty and $I(J \neq \emptyset) = 0$. This is similar to the case of $\mathcal{L}_{Val} = 0$. So $\boldsymbol{d}^*$ becomes a descent direction. $\qquad \square$

## 4.2 Derivations for Eq. (4)

Our derivations follow a standard Expectation-Maximization process. Eq. (3) is actually the E-step to estimate the hidden variable, class label, on the target domain. The detailed derivation is in the following.

$$
\begin{aligned}
p(y = k \mid z = d, \boldsymbol{x}, \boldsymbol{\theta}, \boldsymbol{\phi}_c) &= \frac{p(y = k, z = d \mid \boldsymbol{x}, \boldsymbol{\theta}, \boldsymbol{\phi}_c)}{p(z = d \mid \boldsymbol{x}, \boldsymbol{\theta}, \boldsymbol{\phi}_c)} \\
&= \frac{p(y = k, z = d \mid \boldsymbol{x}, \boldsymbol{\theta}, \boldsymbol{\phi}_c)}{\sum_{k'} p(y = k', z = d \mid \boldsymbol{x}, \boldsymbol{\theta}, \boldsymbol{\phi}_c)} \\
&= \frac{p(z = d \mid y = k, \boldsymbol{x}, \boldsymbol{\theta}, \boldsymbol{\phi}_c) p(y = k \mid \boldsymbol{x}, \boldsymbol{\theta}, \boldsymbol{\phi}_c)}{\sum_{k'} p(z = d \mid y = k', \boldsymbol{x}, \boldsymbol{\theta}, \boldsymbol{\phi}_c) p(y = k' \mid \boldsymbol{x}, \boldsymbol{\theta}, \boldsymbol{\phi}_c)} \\
&= \frac{p(z = d \mid \boldsymbol{x}, \boldsymbol{v}_k) p(y = k \mid \boldsymbol{x}, \boldsymbol{\theta}, \boldsymbol{\phi}_c)}{\sum_{k'} p(z = d \mid \boldsymbol{x}, \boldsymbol{v}_{k'}) p(y = k' \mid \boldsymbol{x}, \boldsymbol{\theta}, \boldsymbol{\phi}_c)} = \rho_{k|d}.
\end{aligned}
$$

And Eq. (4) is a part of the M-step.

For the M-step, we maximize the joint likelihood,

$$
\begin{aligned}
&\prod_{d=1}^{D} \prod_{k=1}^{K} p(y = k, z = d \mid \boldsymbol{x}, \boldsymbol{\theta}, \boldsymbol{\phi}_c)^{I(y=k, z=d)} \\
&= \prod_{d=1}^{D} \prod_{k=1}^{K} p(z = d \mid y = k, \boldsymbol{x}, \boldsymbol{\theta}, \boldsymbol{\phi}_c)^{I(z=d)} p(y = k \mid \boldsymbol{x}, \boldsymbol{\theta}, \boldsymbol{\phi}_c)^{I(y=k)} \\
&= \prod_{d=1}^{D} \prod_{k=1}^{K} p(z = d \mid \boldsymbol{x}, \boldsymbol{v}_k)^{I(z=d)} p(y = k \mid \boldsymbol{x}, \boldsymbol{\theta}, \boldsymbol{\phi}_c)^{I(y=k)}.
\end{aligned}
$$

Then the log likelihood is

$$
\sum_{d=1}^{D} \sum_{k=1}^{K} I(z = d) I(y = k) \log[p(z = d \mid \boldsymbol{x}, \boldsymbol{v}_k) p(y = k \mid \boldsymbol{x}, \boldsymbol{\theta}, \boldsymbol{\phi}_c)].
$$

For target domains where class labels are not available, we replace $I(y = k)$ with the conditional estimation $\rho_{k|d}$. Then we have the final log likelihood:

$$
\sum_{d=1}^{D} \sum_{k=1}^{K} I(z = d) s_{k|d} \log[p(z = d \mid \boldsymbol{x}, \boldsymbol{v}_k) p(y = k \mid \boldsymbol{x}, \boldsymbol{\theta}, \boldsymbol{\phi}_c)],
$$

where $s_{k|d} = I(y = k)$ is for the source domain, and $s_{k|d} = \rho_{k|d}$ is for the target domain.

We decompose the final log likelihood as

$$\sum_{d=1}^{D}\sum_{k=1}^{K} I(z=d)s_{k|d}\log p(z=d \mid \boldsymbol{x},\boldsymbol{v}_k) + \sum_{d=1}^{D}\sum_{k=1}^{K} I(z=d)s_{k|d}\log p(y=k \mid \boldsymbol{x},\boldsymbol{\theta},\boldsymbol{\phi}_c)$$

$$=\sum_{k=1}^{K}\sum_{d=1}^{D} s_{k|d}I(z=d)\log p(z=d \mid \boldsymbol{x},\boldsymbol{v}_k) + \sum_{d=1}^{D} I(z=d)\sum_{k=1}^{K} s_{k|d}\log p(y=k \mid \boldsymbol{x},\boldsymbol{\theta},\boldsymbol{\phi}_c),$$

where the first term is Eq. (4) and learns domain discrimination for each class with a weighted loss (the weight is $s_{k|d}$), and the second term learns classification for each domain. For the source domain, the class labels are hard. Whereas for the target domain, the class labels are soft.

### 4.3 Existence Guarantee for Pareto optimal solution

Similar as in [11], we provide a theoretical guarantee for the existence of a Pareto critical (also named as local Pareto optimal in [15]) solution of our method, where a solution is called Pareto critical if no other solution in its neighborhood can dominate this solution. Specifically, we show by Theorem 1 below that our method will optimize until the solution is Pareto critical and satisfies $\mathcal{L}_{Val}=0$.

**Theorem 1.** *Let $\hat{\boldsymbol{\theta}}$ be the final output model parameters of our method. Let $\boldsymbol{w}^*$ be the solution of the problem in Eq. (6) based on $\hat{\boldsymbol{\theta}}$, and $\boldsymbol{d}^* = \boldsymbol{G}\boldsymbol{w}^*$ be the resulted update direction. We have:*

- *if $\hat{\boldsymbol{\theta}}$ is not Pareto critical, then $\boldsymbol{d}^* \neq \mathbf{0} \in \mathbb{R}^d$;*

- *if $\hat{\boldsymbol{\theta}}$ is Pareto critical, we further consider two cases:*

  - *If $\mathcal{L}_{Val}=0$, then $\boldsymbol{d}^* = \mathbf{0} \in \mathbb{R}^d$ or $(\boldsymbol{d}^*)^T\boldsymbol{G} = \mathbf{0} \in \mathbb{R}^3$. In either case, the update is meaningless and will halt.*
  - *If $\mathcal{L}_{Val}>0$, assuming the gradient of $\mathcal{L}_T$ is consistent with the gradient of $\mathcal{L}_{Val}$, i.e., $\boldsymbol{g}_3^T\hat{\boldsymbol{g}}_v > 0$, then $\boldsymbol{d}^* \neq \mathbf{0} \in \mathbb{R}^d$.*

    *Note that the assumption that $\boldsymbol{g}_3^T\hat{\boldsymbol{g}}_v > 0$ for a Pareto critical $\hat{\boldsymbol{\theta}}$ is reasonable, because when $\hat{\boldsymbol{\theta}}$ is Pareto critical, the model is usually well-trained, and $\boldsymbol{g}_3, \hat{\boldsymbol{g}}_v$ are gradients of the same loss on i.i.d sampled two datasets from the target domain. Then it is reasonable to assume the gradients are consistent.*

The proof of Theorem 1 is as following:

*Proof.*

- If $\hat{\boldsymbol{\theta}}$ is not Pareto critical, we suppose $\boldsymbol{d}^* = \boldsymbol{G}\boldsymbol{w}^* = \mathbf{0} \in \mathbb{R}^d$, where $\boldsymbol{w}_j^* > 0$ for all $j \in \{1,2,3\}$. Since $\hat{\boldsymbol{\theta}}$ is not Pareto critical, there exists a $\boldsymbol{d}' \neq \mathbf{0} \in \mathbb{R}^d$ that satisfies $(\boldsymbol{d}')^T\boldsymbol{g}_j > 0$, for some $j \in \{1,2,3\}$; and $(\boldsymbol{d}')^T\boldsymbol{g}_i \geq 0$, for other $i \in \{1,2,3\}$ and $i \neq j$. Since $\boldsymbol{G}\boldsymbol{w}^* = \mathbf{0}$, then $\boldsymbol{w}_j^*\boldsymbol{g}_j = -\sum_{i\neq j}\boldsymbol{w}_i^*\boldsymbol{g}_i$. Also, because $(\boldsymbol{d}')^T\boldsymbol{g}_j > 0$, then $(\boldsymbol{d}')^T(\boldsymbol{w}_j^*\boldsymbol{g}_j) > 0$ and $(\boldsymbol{d}')^T(\sum_{i\neq j}\boldsymbol{w}_i^*\boldsymbol{g}_i) < 0$. However, $(\boldsymbol{d}')^T(\sum_{i\neq j}\boldsymbol{w}_i^*\boldsymbol{g}_i) < 0$ is not possible because $(\boldsymbol{d}')^T\boldsymbol{g}_i \geq 0$, for $i \in 1,2,3$ and $i \neq j$ and $\boldsymbol{w}_i^* > 0$ for all $i \in 1,2,3$, which generates a contradiction for $\boldsymbol{d}^* = \mathbf{0} \in \mathbb{R}^d$.

- If $\hat{\boldsymbol{\theta}}$ is Pareto critical, we consider two cases.

  - If $\mathcal{L}_{Val}=0$, then from Theorem 1 we know that $\boldsymbol{d}^*$ becomes a descent direction, i.e., $(\boldsymbol{d}^*)^T\boldsymbol{g}_j \geq 0, \ \forall j \in 1,2,3$. Since $\hat{\boldsymbol{\theta}}$ is Pareto critical, then there does not exist $j \in 1,2,3$ such that $(\boldsymbol{d}^*)^T\boldsymbol{g}_j > 0$. Then $(\boldsymbol{d}^*)^T\boldsymbol{g}_j = 0, \ \forall j \in 1,2,3$. Thus, we either have $\boldsymbol{d}^* = \mathbf{0} \in \mathbb{R}^d$ or just $(\boldsymbol{d}^*)^T\boldsymbol{G} = \mathbf{0} \in \mathbb{R}^3$.

  - Consider the second case, $\mathcal{L}_{Val}>0$, then $\hat{\boldsymbol{g}}_v \neq \mathbf{0} \in \mathbb{R}^d$. Given the assumption that $\boldsymbol{g}_3^T\hat{\boldsymbol{g}}_v > 0$, then $\boldsymbol{g}_3 \neq \mathbf{0} \in \mathbb{R}^d$. Then $\boldsymbol{w}'$ can be $[0,0,1]$ and $\boldsymbol{\gamma}' = (\boldsymbol{G}\boldsymbol{w}')^T\hat{\boldsymbol{g}}_v = \boldsymbol{g}_3^T\hat{\boldsymbol{g}}_v > 0$. Therefore, the maximized $\boldsymbol{\gamma}^* \geq \boldsymbol{\gamma}' > 0$, then from Theorem 1 we know that $(\boldsymbol{d}^*)^T\hat{\boldsymbol{g}}_v > 0$, which means $\boldsymbol{d}^* \neq \mathbf{0} \in \mathbb{R}^d$.

$\square$

In summary, we can obtain a Pareto critical solution with simple iterative gradient-based update rule $\boldsymbol{\theta}_{t+1} = \boldsymbol{\theta}_t + \eta\boldsymbol{d}$.