# OpenReview forum: "Pareto Domain Adaptation"
_NeurIPS.cc/2021/Conference — NeurIPS 2021 Poster_

### Official Review · Reviewer_M9tJ · 2021-07-15

**Rating:** 6
**Confidence:** 3

**Summary:**

The paper considers the common class of domain adaptation algorithms consisting in minimizing the sum of a source risk term and a divergence between the domains. Since both terms are required to be small, the authors examine the tradeoff between them via their associated Pareto front. This latter having non-convex regions non-reachable by the classic weighted sum minimization procedures, a multi-objective optimization procedure is proposed in order to reach all of the regions of the Pareto front and to access tradeoffs that are not accessible by the classic approach. Specifically, the authors use a modified version of a multi-objective optimization procedure, where they define a guidance direction that adjusts the directions of both gradients of the two cost function terms. The guidance direction is the gradient of a surrogate loss assessing performance on the target domain. To show the empirical performance of their idea, the authors present experiments in which their method improves previously proposed DA algorithms.

**Limitations And Societal Impact:**

One limitation is stated in the conclusion, which is the restriction to the traditional domain adaptation setting. It would be clearer to specify what is meant by traditional (single-sourced, closed set, etc). The societal impacts are discussed in the supplementary material.

**Main Review:**

The paper is clearly written and easy to follow. The idea is an interesting enhancement for DA methods, and the explanations about multi-objective optimization are succinct.

## Strengths

* Bringing multi-objective optimization to domain adaptation, which is an extension to several proposed algorithms.
* Utilizing the domain discriminator to construct a surrogate for the target domain loss, an original idea that goes beyond the classic role of aligning the two domains.
* The adaptation of the idea from EPO [6] to the specific case of domain adaptation by modifying the guidance direction ${\bf d}_{bal}$.

## Weaknesses:

* Although soft-labels are used for the target domain, no comparison to methods using pseudo-labels is used.
* Not mentioning of the link to methods performing class-wise domain alignment after the optimization problem of Equation (4) (e.g. [5])
* Related Works section:
  * It is necessary to mention theoretical DA contributions, at least the seminal ones (e.g. [1,2]). Indeed, the idea of the domain discriminator introduced in [1] has inspired [3], in addition to the more recent GAN paper [4].
* I believe that there is a problem in the proof of Theorem 1, specifically for the case $\gamma^{\*}\leq 0$ (lines 54 to 57 in the supplementary material). Indeed, we have $({\bf G}{\bf w}^*)^T{\bf g_v} \leq 0 \Leftrightarrow \sum_{i=1}^3w_i{{\bf g}_i^T{\bf g_v}}\leq 0$. This implies that there is some $j \in \{1,2,3\}$ such that   $w_j{{\bf g}_j^T{\bf g_v}}\leq 0$, which further implies that  ${{\bf g}_j^T{\bf g_v}}\leq 0$ if $w_j>0$. This, however, does not imply that $\forall i \in {1,2,3}, {\bf g}_i^T{\bf g}_v < 0$ (contradicting the statement made in line 55 " there is no gradient ${\bf g}_j$, $j \in  \{1, 2, 3\}$, for which ${\bf g}_j^T {\bf g}_v \geq 0$").
* Lacking notation definitions:
  * $\mathcal L$ (line 87) is not defined.
  * ${\mathit{ v}}_{\mathbf c}$ (Equation set (3)), and $\lambda_0, \lambda_1$ (line 338) are not defined
  * In Equation (5), $X_t$ is not defined. Is it the finite set of observed target instances ?
  * The definition of gradients ${\bf g}_j$ is given in line 222, after their first mention in line 208.
  * In Equation (4), either a $\max$ should be instead of $\min$, or a minus sign has to be added, as the expression is a log-likelihood to be maximized.
  * In the objective functions in (6), $\bf 1$ should be defined.

## Comments

* Line 173 "The probability of the i-th ..." --> "The class-conditional probability of the i-th ..."

### Minor comments:

* Line 151 ". While" --> Either ", while", or ". Meanwhile/On the other hand...".
* Line 215: "Chapter" --> "Section"
* Lines 181-182:
  * "containing" --> "contain" (otherwise the sentence seems incomplete).
  * "belongs" --> "belong"
* Supplementary, line 53: "then $\gamma^*$" --> "then $\gamma^*>0$".
* Line 252: "$n\gg m$" --> "$d\gg m$" ?

## Suggestions

* The "soft labels" term for the target domain, mentioned in the supplementary (line 69), offers more intuition about the objective in Equation (4). It should be mentioned in the main paper.
* I recommend performing the experiments from Figure 3 to other datasets (toy 2d data) .
* The last paragraph title should contain "multi-objective", as "Gradient-based optimization" refers to mono-objective first order methods, as multi-objective optimization is not yet commonly adopted by the ML community.

* Although its meaning is given in line 219, the notation ${\bf g}_v$ can be confused for the column vector of $\bf G$ that has index $v$. I suggest using another letter for denoting the guidance direction.

## Questions:

* Is the extension to regression immediate ?
* Can the surrogate target loss be used to select hyper-parameters in an unsupervised way for other DA methods ?
* Isn't the condition  $\mathcal L_{val}>0$ too strict in the definition of optimization problem (6) ? Is it beneficial to relax it ? (e.g. $\mathcal L_{val}>\varepsilon$ and $\mathcal L_{val}<\varepsilon$) for a given small $\varepsilon$ ?
* Have you tried to visualize the Pareto front for different domain gaps ? It is interesting to see how it evolves as the gap between the two domains becomes larger.

## References

[1] Ben-David, S., Blitzer, J., Crammer, K. et al. A theory of learning from different domains. Mach Learn 79, 151–175 (2010).

[2] Mansour, Y., Mohri, M., Ro, J., Theertha Suresh, A. &amp; Wu, K.. (2021).  A Theory of Multiple-Source Adaptation with Limited Target Labeled Data . <i>Proceedings of The 24th International Conference on Artificial Intelligence and Statistics</i>, in <i>Proceedings of Machine Learning Research</i> 130:2332-2340

[3] Ganin, Y., & Lempitsky, V. (2015, June). Unsupervised domain adaptation by backpropagation. In International conference on machine learning (pp. 1180-1189). PMLR.

[4] Goodfellow, I., Pouget-Abadie, J., Mirza, M., Xu, B., Warde-Farley, D., Ozair, S., ... & Bengio, Y. (2014). Generative adversarial nets. Advances in neural information processing systems, 27.

[5] Long, M., Cao, Z., Wang, J., & Jordan, M. I. (2018). Conditional Adversarial Domain Adaptation. *Neural Information Processing Systems*.

[6] Mahapatra, D., & Rajan, V. (2020, November). Multi-task learning with user preferences: Gradient descent with controlled ascent in pareto optimization. In *International Conference on Machine Learning* (pp. 6597-6607). PMLR.

---------
## Update after authors rebuttal
I thank the authors for their rebuttal that clarified the different concerns I raised.

**Time Spent Reviewing:**

4

---

> ### Author Response · Authors · 2021-08-10
> **Response to Reviewer M9tJ (PART I)**
>
> Thank you for your valuable feedback and meticulous comments, and for recognizing the contributions in our work. We will incorporate your suggestions in the revision.
> We address your concerns and questions below. Please let us know if there are any further questions.
>
> ----------------------------------------------------------------------Weaknesses------------------------------------------------------------------------\
> **Q1:** Although soft-labels are used for the target domain, no comparison to methods using pseudo-labels is used.
>
> **A1:**
>  The major difference between our method and other methods that use pseudo-labels is that ours adopt soft labels to guide the overall gradient descent direction of training objectives. Specifically, we refine the target soft labels by introducing extra information, i.e., domain labels. Then, we adopt the refined soft labels to construct the CIM loss for guiding the overall descent direction of training objectives. On the other hand, several works derive pseudo labels from the soft labels, which will be used to directly train the networks.
>
>  Below is the comparison between our ParetoDA and some DA methods using pseudo-labels on Office-31, which further verify the effectiveness and superiority of our method.
>
> | Method| A:W    | D:W    | W:D     | A:D     | D:A     | W:A    |  Avg |
> | ----  | ----   | ----   | ----    | -----   | ----    | ----   | ---- |
> |PCDA[1]|92.5±0.3|98.7±0.2|100.0±0.0|93.0±0.2 |73.5± 0.3|72.5±0.3| 88.3 |
> |SPL[2] |92.7    |  98.7  | 99.8    |   93.0  | 76.4    | 76.8   | 89.6 |
> |DCP[3] |95.3±0.1|98.3±0.4|100.0±0.0| 91.6±0.2|73.1±0.2 |72.7±0.3| 88.5 |
> |DANN + ParetoDA|95.5 ± 0.1 |98.7 ± 0.3| 100.0 ± 0.0 |93.8 ±0.2| 76.7±0.2 |76.3 ±0.3 |90.2|
> CDAN + ParetoDA |95.0 ± 0.2 |98.9±0.1 |100.0 ±0.0 |95.4 ± 0.2| 77.6 ± 0.3| 75.7 ± 0.2| 90.4|
> MDD + ParetoDA  |95.4 ± 0.2 |98.9 ± 0.2 |100.0 ± 0.0| 94.4 ± 0.3| 76.2 ± 0.4| 75.8 ± 0.2| 90.1|
>
> > [1] J. C., M.J., T.K., C.K. (2019). Pseudo-Labeling Curriculum for Unsupervised Domain Adaptation. BMVA, pp.67.\
> [2] Wang, P. B. (2020). Unsupervised Domain Adaptation via Structured Prediction Based Selective
>                Pseudo-Labeling. AAAI, 6243-6250.\
> [3] Chen, L. L., Chen, Lin. (2021). Unsupervised domain adaptation via double classifiers based on high
>                confidence pseudo label. CoRR, abs/2105.04729.
>
> **Q2:** Not mentioning of the link to methods performing class-wise domain alignment after the optimization problem of Equation (4) (e.g. [5])
>
> **A2:** CDAN [5] integrates the classifier predictions to the domain discriminator to perform class-wise domain alignment via adversarially training the feature extractor and the domain discriminator. While in ParetoDA, we optimize Eq. (4) to train a well-performed domain discriminator for each class, which is orthogonal to the training process in previous methods, such as CDAN [5]. By doing this, we can leverage those class-wise domain discriminators to model $p(z=d|\boldsymbol{x},\boldsymbol{v_c})$, which will be used in Eq. (3) to refine the target predictions $p(y=c|z=d,\boldsymbol{x},\boldsymbol{\theta},\boldsymbol{\phi}_c)$ by introducing the domain labels.
>
> We will state the link between our method and other domain alignment methods in the final version.
>
> **Q3:** Related Works section: It is necessary to mention theoretical DA contributions, at least the seminal ones (e.g. [1,2]). Indeed, the idea of the domain discriminator introduced in [1] has inspired [3], in addition to the more recent GAN paper [4].
>
> **A3:** Thanks for your thoughtful suggestion. Indeed, [1, 2] provide important theoretical insights for the domain adaptation community.  Inspired by [1], many works (e.g., [3]) design methods aiming to lower the upper bound of generalization error via various mechanisms. We will discuss the impact of [1, 2] and the relatedness between our work and them in the Related Work section.
>
> Nonetheless, we would like to point out that our domain discriminators are for enhancing the prediction for target domains, unlike that in [3], their domain discriminator is for learning domain-invariant features. Therefore, our proposed technique is still novel.
>
> **Q4:** I believe that there is a problem in the proof of Theorem 1, specifically for the case  $\gamma^* \le 0$.
>
> **A4:**: Thank you for your careful review. We would like to point out that $\gamma^*=(Gw^*)^Tg_v$ is already maximized, which means that if there is at least one $g_j$, supposing $j=1$, that satisfies $g_j^Tg_v>0$, then $w^*$ can just be $[1,0,0]^T$ and we can get $\gamma^*=(Gw^*)^Tg_v=([g_1,g_2,g_3][1,0,0]^T)^Tg_v=g_1^Tg_v>0$. So if the maximized $\gamma^*\leq0$, it means that we cannot find even one g_j that satisfies $g_j^Tg_v>0$. Therefore, it means that there is no gradient $g_j, j\in\{1,2,3\}$, for which $g_j^Tg_v>0$. Nonetheless, we admit that the original presentation, “for which $g_j^Tg_v\geq0$”, has a typo. It should be “for which $g_j^Tg_v>0$”.
>
>
> **Q5:** Lacking notation definitions:
>
> **A5:** Thanks for pointing this out, we will make this clear in the final version.
>
> - $\mathcal{L}$ is not defined
>   - $\mathcal{L}$ denotes a non-negative loss function.
> - $v_{c}$ (Equation (3)), and $\lambda_0$ and $\lambda_1$ (line 338) are not defined
>   - $v_c$ denotes the parameters of $c$-th domain discriminator corresponding to class $c$, which aims to distinguish the domain label of the samples from class $c$.  $\lambda_0$ and $\lambda_1$ are the scale factors for the object classification objective and domain alignment objective, respectively. We adopt the $\lambda_0$ and $\lambda_1$ to vary the value scale of training objectives to test whether ParetoDA is sensitive to it.
> - In Equation (5), $X_t$ is not defined. Is it the finite set of observed target instance?
>   - $X_t$ is the finite set of observed target instances. We will substitute $X_t$ with $D_t$ to maintain consistency.
> - The definition of gradients $g_i$ is given in line 222, after their first mention line 208
>   - We will define the $g_i$ in line 208.
> - In Equation (4), either a $\max$ should be instead of $\min$, or a minus sign has to be added as the expression is a log-likelihood to be maximized.
>   - We will add a minus sign to make this correct. Thanks for carefully reading our manuscript.
> - In the objective function in (6), $\boldsymbol{1}$ should be defined
>   - $\boldsymbol{1}$ a matrix whose elements are all $1 \in R$.
>
> ------------------------------------------------------------------------Comments--------------------------------------------------------------------------\
> **Q6:** Line 173 "The probability of the i-th ... " --> "The class-conditional probability of the i-th ..."
>
> **A6:** Thanks for your insightful suggestion.  We will revise it as "The domain-condition classification probability of the i-th ..." in the final version.
>
> **Q7:** Line 151 "While" --> "Either"; Line 215 "Chapter" --> "Section"; Line 181-182: "containing" --> "contain" (otherwise the sentence seems incomplete); "belongs" --> "belong"; Supplementary, line 53: "then $\gamma^*$" --> then $ \gamma^* > 0$; "Line 252: "n $\gg$ m" --> "d $\gg$ m"?
>
> **A7:** Thanks for your suggestions, we will revise our manuscript thoroughly.

---

> ### Author Response · Authors · 2021-08-10
> **Response to Reviewer M9tJ (PART II)**
>
> --------------------------------------------------------------------Suggestions----------------------------------------------------------------------\
> **Q8:** The "soft labels" term for the target domain, mentioned in the supplementary (line 69), offers more intuition about the objective in Equation (4). It should be mentioned in the main paper.
>
> **A8:** Thanks for your insightful suggestion. Indeed, it would help readers to understand better if we describe the $s_{c|d}=\rho_{c|d}$ as the soft labels for target domain produced by the classifier. We will make this clear in the final version.
>
> **Q9:** I recommend performing the experiments from Figure 3 to other datasets (toy 2d data) .
>
> **A9:**
> Thanks for your insightful comment. It is unfortunate that the toy examples conducted by previous MOO methods (e.g., EPO and ParetoDA) can't apply to our ParetoDA due to the unique setting of DA. In standard MOO methods, the toy example can be simply designed by designing two objective functions and the data are drawn from the same distribution. On the contrary, ParetoDA is designed for DA setting, where the objective losses are source classification loss and domain alignment loss. Unlike the relationship between two losses in standard MOO problems which can be designed directly, the relationship between source classification loss and domain alignment loss is difficult to design in prior. What's more, the data in DA are sampled from different domains.
>
> In such a case, we perform the experiment in Fig. 3 on a toy 2D dataset, i.e., inter twining moons problems proposed by [1]. Specifically, we generate two classes of source samples, one is an upper moon, the other is a lower moon, labeled 0 and 1, respectively. For the target samples, we enhance domain gap by rotating 30 degrees of the distribution of the source samples to generate them. We generate 100 samples per class as the training samples. Both feature generator and classifiers employ a 3-layered fully-connected network.
>
> The analytical experiments are still conducted based on DANN. And the visualization results can be seen in the following link: [https://anonymous.4open.science/r/NIPS2021_rebuttal-Paper1631/Fig_optimization_visualization_toy.pdf]
>
> It can be seen that our ParetoDA still attain the Pareto optimal solution closest to the ideal one, which verifies the effectiveness of our method.
>
> > [1] Pedregosa, F.; Varoquaux, G.; Gramfort, A.; Michel, V.;
> Thirion, B.; Grisel, O.; Blondel, M.; Prettenhofer, P.; Weiss,
> R.; Dubourg, V.; et al. 2011. Scikit-learn: Machine Learning
> in Python. JMLR 12: 2825–2830.
>
>
> **Q10:** The last paragraph title should contain "multi-objective", as "Gradient-based optimization" refers to mono-objective first order methods, as multi-objective optimization is not yet commonly adopted by the ML community.
>
> **A10:** Thanks for your insightful advice. To help readers understand better for our method, we will change the title to "Multi-objective gradient-based optimization" in the final version.
>
> **Q11:** Although its meaning is given in line 219, the notation $\boldsymbol{g}_v$, can be confused for the column vector of $\boldsymbol{G}$ that has index $v$. I suggest using another letter for denoting the guidance direction.\
> **A11:** Thanks for your suggestion. To avoid confusion, we will use a new symbol for the gradient descent direction to replace the old one $\boldsymbol{g}_v$.
>
> ---------------------------------------------------------------------Questions-----------------------------------------------------------------------\
> **Q12:** Is the extension to regression immediately?
>
> **A12:** For regression problems, we still can apply ParetoDA to search for ideal Pareto optimal solution.
>
>   The major difference between regression and classification is that the label of regression tasks is continuous while that of classification is discrete.
>
>   Nonetheless, ParetoDA is designed to mitigate the conflict of multiple training objectives. Thus, in the context of regression tasks (e.g., [4]), we can still adopt PraretoDA to control the overall gradient descent direction and coordinately optimize all training objectives. It is worth noting, however, due to the continuous labels which are different from classification task, it needs to redesign the surrogate loss to replace CIM tailored for classification tasks.
>
> > [4] Chen, Xinyang, et al. "Representation Subspace Distance for Domain Adaptation Regression." *International Conference on Machine Learning*. PMLR, 2021.
>
> **Q13:** Can the surrogate target loss be used to select hyper-parameters in an unsupervised way for other DA methods?
>
> **A13:** We believe that it's promising to adopt the proposed CIM loss to guide the selection for hyper-parameters.
>
>  To refine the target predictions, we introduce additional domain labels based on Bayesian theory, which is more accurate compared to the original version. Moreover, we adopt information maximization onto the improved target predictions to make them individually certain and globally diverse, which with the same properties as the output obtained by optimizing the supervised target classification loss.
>
>  Thus, our surrogate loss has the ability to well mimic the real supervised target classification loss on validation set, and then can serve as a metric to select hyper-parameters in an unsupervised way for other DA methods.
>
>  We will further explore the potential of CIM loss in this field in the following works.
>
>
> **Q14:** Isn't the condition $L_{Val} > 0$ too strict in the definition of optimization problem (6) ? Is it beneficial to relax it ? (e.g. $L_{Val} > \epsilon $ and $L_{Val} < \epsilon$) for a given small $\epsilon$?
>
> **A14:** Thanks for your constructive suggestion. Adding a small $\epsilon > 0$ to relax the condition $L_{Val} > 0 $ and $L_{Val} =0$ is indeed beneficial. Since when $L_{Val}$ is less than a small value, it has trivial guiding significance. We have found this point during experiment and set $\epsilon$ as 1e-3 in practice. However, for clarity and simplicity, we did not include the relaxation variable $\epsilon$ in our formula. We will revise the formula as suggested in the final revision.
>
>
> **Q15:** Have you tried to visualize the Pareto front for different domain gaps? It is interesting to see how it evolves as the gap between the two domains becomes larger.
>
> **A15:**
> To build domain pairs with different gaps, we adopt stylegan [1] to generate images with different gaps.
>
> Specifically, we first train the stylegan model on the Office-31 dataset and the feature extractor (e.g., a ResNet-50 model) using DANN on task Webcam $\rightarrow$ Amazon. After the two modules are well trained separately, we freeze them during the following phases. Second, we interpolate the source features towards target domain in the feature space, since the feature representations contain more semantic information than pixels. By doing so, we can obtain generated images with target styles. Via tuning the strength of interpolation, we can generate images with different domain gaps.
>
> Hence we have three domain pairs: the default one, the one with medium domain gap, and the one with small domain gap. The example images are shown in [https://anonymous.4open.science/r/NIPS2021_rebuttal-Paper1631/Fig_domain_gaps.pdf]. One can observe that source image has complex real-world background, while the target image has simple and blank background. In addition, the generated image has simpler background compared with source image.
>
> we train the model with these generated domains and then visualize the Pareto front for their different gaps, which can be seen in [https://anonymous.4open.science/r/NIPS2021_rebuttal-Paper1631/Fig_pareto_front_domain_gaps.pdf]. One can observe that the smaller the domain gap is, the smaller the conflict between the objectives is. This might because as the domain gap decreases, the model becomes more capable to better achieve source domain classification and domain alignment at the same time.
>
> [1] Karras, Tero, et al. "Analyzing and improving the image quality of stylegan." *Proceedings of the IEEE/CVF Conference on Computer Vision and Pattern Recognition*. 2020.

---

> > ### Comment · Reviewer_M9tJ · 2021-08-18
> > **Concerns addressed, score increased**
> >
> > I would like to thank the authors for their detailed rebuttal, in which they clarified the different points I raised in my review. Due to these clarifications, I update my score to 6.

---

> > > ### Author Response · Authors · 2021-08-18
> > > **Thank You & Authors' Response**
> > >
> > > Thank you for reading our response and increasing the rating!
> > >
> > > We will carefully revise the manuscript following your kind comments.
> > >
> > > Thanks again for your valuable comments and suggestions!

---

### Official Review · Reviewer_MkjU · 2021-07-16

**Rating:** 6
**Confidence:** 3

**Summary:**

This paper represents a domain adaptation (DA) problem in a multi-objective optimization scheme, which can be solved by a gradient-based perspective. The main contributions of the paper include:

1. Proposes a Pareto Domain Adaptation (ParetoDA) approach to control the overall optimization direction, aiming to optimize all training objectives. That ParetoDA scheme can be plugged into various DA methods and enhance their performance.
2. To find a desired Pareto optimal solution, the authors design a surrogate objective function that mimics the target classification task.
3. Provides experimental results that show the effectiveness of ParetoDA compared to some DA methods.

**Ethics Review Area:**

["I don’t know"]

**Limitations And Societal Impact:**

Yes

**Main Review:**

Quality: The idea of representing the domain adaptation setting to a multi-objective optimization scheme is interesting. However,  I'm a bit concern about the theoretical contributions of the paper since the method is quite straightforward. Moreover, there is no theoretical guarantee for the existence of a Pareto optimal of the corresponding multi-objective optimization problem by using the proposed ParetoDA. Also, it would be much more interesting if the authors can develop an upper bound for the generalization error of the DA.

Clarify: Here are several critical points:
- It's unclear for me about the formulation of the loss $\mathcal L_T$ in (5). Why minimizing that loss can handle the domain-shift in the domain adaptation problem? That's important since it can verify the main contribution of the paper.
- (Line 218) It's unclear how to select the validation set $X_t^{Val}$.
- The linear programming (6) misses the condition $\sum w=1$? Does this affect the solution of (6), and consequently the performance of the DA framework?
Originality: To the best of my knowledge, this paper is the first work that introduces the use of multi-objective optimization in a domain adaptation problem.

Significance: Based on my above comments and evaluations, I personally think that it's a bit below the acceptance rate of a NeurIPS publication.


**Time Spent Reviewing:**

7

---

> ### Author Response · Authors · 2021-08-10
> **Response to Reviewer MkjU (PART I)**
>
> Thank you for your valuable comments, and for recognizing the contributions in our work. We will incorporate your suggestions in the revision.
> We address your concerns and questions below. Please let us know if there are any further questions.
>
> ------------------------------------------------------------------------Quality--------------------------------------------------------------------------\
> **Q1:** There is no theoretical guarantee for the existence of a Pareto optimal of the corresponding multi-objective optimization problem by using the proposed ParetoDA.
>
> **R1:**
> Thanks for your suggestion. Below we provide the theoretical guarantee for the existence of a Pareto optimal of the corresponding multi-objective optimization problem by using the proposed ParetoDA.
>
> Similar as in [1], we provide a theoretical guarantee for the existence of a Pareto critical (also named as local Pareto optimal in [2]) solution of our method, where a solution is called Pareto critical if no other solution in its neighborhood can dominate this solution. Specifically, we show by Theorem 2 below that our method will optimize until the solution is Pareto critical and satisfies $\mathcal{L}_{Val}=0$.
>
> > [1] Lin, X., Zhen, H. L., Li, Z., Zhang, Q. F., & Kwong, S. (2019). Pareto multi-task learning. Advances in neural information processing systems, 32, 12060-12070.\
> [2] Mahapatra, D., & Rajan, V. (2020, November). Multi-task learning with user preferences: Gradient descent with controlled ascent in pareto optimization. In International Conference on Machine Learning (pp. 6597-6607). PMLR.
>
> **Theorem 2.**
>
> Let $\hat{\boldsymbol{\theta}}$ be the final output model parameters of our method. Let $\boldsymbol{w}^*$ be the solution of the problem in Eq. (6) based on $\hat{\boldsymbol{\theta}}$, and $\boldsymbol{d}^*=\boldsymbol{G}\boldsymbol{w}^*$ be the resulted update direction. We have:
>
> - if $\hat{\boldsymbol{\theta}}$ is not Pareto critical, then $\boldsymbol{d}^*\neq\boldsymbol{0}\in\mathbb{R}^d$;
>
> - if $\hat{\boldsymbol{\theta}}$ is Pareto critical, we further consider two cases:
>
>     1) If $\mathcal{L}_{Val}=0$, then $\boldsymbol{d}^*=\boldsymbol{0}\in\mathbb{R}^d$ or $(\boldsymbol{d}^*)^T\boldsymbol{G}=\boldsymbol{0}\in\mathbb{R}^3$. In either case, the update is meaningless and will halt.
>
>     2) If $\mathcal{L}_{Val}>0$, assuming the gradient of $\mathcal{L}_T$
>
>      is consistent with the gradient of $\mathcal{L}_{Val}$, i.e., $\boldsymbol{g}_3^T\boldsymbol{g}_v>0$,
> then  $\boldsymbol{d}^*\neq\boldsymbol{0}\in\mathbb{R}^d$.
>
> Note that the assumption that $\boldsymbol{g}_3^T\boldsymbol{g}_v>0$ for a Pareto critical $\hat{\boldsymbol{\theta}}$ is reasonable, because when $\hat{\boldsymbol{\theta}}$ is Pareto critical, the model is usually well-trained, and $\boldsymbol{g}_3,\boldsymbol{g}_v$ are gradients of the same loss on i.i.d sampled two datasets from the target domain. Then it is reasonable to assume the gradients are consistent.
>
> We prove Theorem 2 in the following.
>
> **Proof.**
>
> - If $\hat{\boldsymbol{\theta}}$ is not Pareto critical, we suppose $\boldsymbol{d}^*=\boldsymbol{G}\boldsymbol{w}^*=\boldsymbol{0}\in\mathbb{R}^d$, where $\boldsymbol{w}_j^*>0$ for all $j\in$ {1,2,3}.
>
>     Since $\hat{\boldsymbol{\theta}}$ is not Pareto critical, there exists a $\boldsymbol{d}'\neq\boldsymbol{0}\in\mathbb{R}^d$
> that satisfies $  (\boldsymbol{d}')^T\boldsymbol{g}_j>0$, for some $j\in$ {1,2,3};
> and $  (\boldsymbol{d}')^T\boldsymbol{g}_i\geq0$, for other $i\in$ {1,2,3} and $i\neq j$.
>
>    Since $\boldsymbol{G}\boldsymbol{w}^*=\boldsymbol{0}$,
> then
> $\boldsymbol{w}_j^* \boldsymbol{g}_j$
>
>     $= - \sum_{i\neq j}\boldsymbol{w}_i^* \boldsymbol{g}_i$ .
>
>     Also, because $  (\boldsymbol{d}')^T\boldsymbol{g}_j>0$, then $  (\boldsymbol{d}')^T(\boldsymbol{w}_j^*\boldsymbol{g}_j)>0$
>
>     and
> $(\boldsymbol{d}')^T(\sum_{i\neq j}\boldsymbol{w}_i^*\boldsymbol{g}_i)<0$.
>
>     However, $(\boldsymbol{d}')^T(\sum_{i\neq j}\boldsymbol{w}_i^*\boldsymbol{g}_i)<0$ is not possible because
> $  (\boldsymbol{d}')^T\boldsymbol{g}_i\geq0$, for $i\in$ {1,2,3} and $i\neq j$
> and
> $\boldsymbol{w}_i^*>0$ for all $i\in$ {1,2,3},
>
>     which generates a contradiction for $\boldsymbol{d}^*=\boldsymbol{0}\in\mathbb{R}^d$.
>
> - If $\hat{\boldsymbol{\theta}}$ is Pareto critical, we consider two cases.
>
>     - If $\mathcal{L}_{Val}=0$, then from Theorem 1 we know that $\boldsymbol{d}^*$ becomes a descent direction, i.e.,
>   $  (\boldsymbol{d}^*)^T\boldsymbol{g}_j\geq 0, \ \forall j\in$ {1,2,3}. \
> Since $\hat{\boldsymbol{\theta}}$ is Pareto critical, then there does not exist $ j\in$ {1,2,3} such that $  (\boldsymbol{d}^*)^T\boldsymbol{g}_j> 0$. Then$ (\boldsymbol{d}^*)^T\boldsymbol{g}_j= 0, \ \forall j\in$ {1,2,3}. \
> Thus, we either have $\boldsymbol{d}^*=\boldsymbol{0}\in\mathbb{R}^d$ or just $(\boldsymbol{d}^*)^T\boldsymbol{G}=\boldsymbol{0}\in\mathbb{R}^3$.
>
>     - Consider the second case, $\mathcal{L}_{Val}>0$, then $\boldsymbol{g}_v\neq\boldsymbol{0}\in\mathbb{R}^d$.\
> Given the assumption that $\boldsymbol{g}_3^T\boldsymbol{g}_v>0$, then $\boldsymbol{g}_3\neq\boldsymbol{0}\in\mathbb{R}^d$.
> Then $\boldsymbol{w}'$ can be $[0,0,1]$ and $\boldsymbol{\gamma}'=(\boldsymbol{G}\boldsymbol{w}')^T\boldsymbol{g}_v=\boldsymbol{g}_3^T\boldsymbol{g}_v>0$.
>
>         Therefore, the maximized $\boldsymbol{\gamma}^*\geq \boldsymbol{\gamma}'>0$,
>
>         then from Theorem 1 we know that $(\boldsymbol{d}^*)^T\boldsymbol{g}_v>0$, which means $\boldsymbol{d}^*\neq\boldsymbol{0}\in\mathbb{R}^d$.
>
> In summary, we can obtain a Pareto critical solution with simple iterative gradient-based update rule $\theta_{t+1} = \theta_t + \eta d$. We will add the above theoretical guarantee to the supplementary materials.
>
> **Q2:** It would be much more interesting if the authors can develop an upper bound for the generalization error of the DA.
>
> **R2:**
> Thanks for your valuable comment. Actually, we have not developed a strict upper bound for the generalization error of the target domain in DA. However, **it can be proved that our ParetoDA can effectively limit the terms of the existing upper bound** proposed by Ben-David et al. [3]. We provide the theoretical derivation process below.
>
> In UDA, the theory proposed by Ben-David et al. [3] provides an upper bound for the generalization error on target samples, which is mainly composed of three terms: 1) the expected error on source domain $\epsilon_S(h)$; 2) the $H\Delta H$ divergence between source and target domain $d_{H\Delta H}(D_s, D_t)$; 3) the combined error $\lambda$ of the ideal joint hypothesis $h^*$. The formulation is as follows:
>
> $\epsilon_T(h) \le \epsilon_S(h) + 1/2 d_{H\Delta H}(D_s, D_t)+ \lambda $;
>
> where $\lambda = \epsilon_S(h^*) + \epsilon_T(h^*)$ and $h^* = \argmin_{h\in H} \epsilon_S(h)+\epsilon_T(h)$.
>
> The goal of the UDA model is to limit the upper bound of the expected error on the target samples to a small value.
> In previous DA methods, the first term, $\epsilon_S(h)$, is expected to be small because reliable labels are owned in the source domain. And the third term $\lambda$ is also generally considered sufficiently small. Thus, most DA methods mainly focus on reduce the second term $d_{H\Delta H}(D_s, D_t)$ by domain alignment.
>
> However, if one of the three terms goes down and another term goes up, then the model still can not effectively reduce the upper bound of $\epsilon_T(h)$. Excessive either source domain classification or domain alignment will cause damage to the other.
>
> In our ParetoDA, by using MOO framework and pareto optimization, **we can simultaneously optimize the source classification loss and the domain alignment loss, so as to make $\epsilon_S(h)$ and $d_{H\Delta H}(D_s, D_t)$ decrease synchronously**. That is, in ParetoDA, the $\epsilon_S(h)$ is further reduced under the premise that this operation does not increase $d_{H\Delta H}(D_s, D_t)$ and vice versa.
>
> Moreover, the training of our proposed CIM loss on target samples and the optimization guidance of the surrogate loss on validation set $L_{Val}$, both contribute to the classification ability of the model to the target domain, and then **reduce the combined error $\lambda$ of the ideal joint hypothesis** $h^*$.
>
> In this way, **the upper bound of $\epsilon_T(h)$ can be effectively reduced in our work**. The above discussion will be added into supplementary materials to verify the effectiveness of our method theoretically.
>
> It is worth noting that in addition to the theoretical support, **the effectiveness of our method has already been strongly supported in the aspect of logic and experiment**.
>
>
> > [3] Shai Ben-David, John Blitzer, Koby Crammer, Alex
> Kulesza, Fernando Pereira, and Jennifer Wortman Vaughan.
> A theory of learning from different domains. In Machine
> learning, volume 79, pages 151–175, 2010. 2

---

> ### Author Response · Authors · 2021-08-10
> **Response to Reviewer MkjU (PART II)**
>
> ------------------------------------------------------------------------Clarity--------------------------------------------------------------------------\
> **Q3:** It's unclear for me about the formulation of the loss $L_T$ in (5). Why minimizing that loss can handle the domain-shift in the domain adaptation problem?
>
> **R3:** There may exist a misunderstanding here: the domain-shift in domain adaptation is handled by the domain alignment loss $L_A$ in the original method rather than $L_T$ in Eq. 5. $L_T$ is designed as a surrogate for the target classification loss and serves as the guidance of our optimization.
>
> Specifically, we first propose a target-prediction enhancing mechanism based on additional domain labels and Bayesian theory ($\rho_{c|d}$ in Eq. 3), then we adopt information maximization on our improved target predictions to make the outputs be individually certain and globally diverse ($L_T$ in Eq. 5), which with the same properties as the output obtained by optimizing the supervised target classification loss.
>
> Thus, $L_T$ is mainly used to mimic the target classification loss, we calculate this loss on the validation set ($L_{Val}$) and then leverage its gradient as the guidance of optimization direction, so as to search for the desirable Pareto optimal solution.
>
>
> **Q4:** (Line 218) It's unclear how to select the validation set $X_t^{Val}$.
>
> **R4:** We split 10% data randomly from the original target set as the validation set, and the rest 90% data are taken as the training set. Note that different from the validation set in traditional setting where the ground truth labels are available, we still don't use any ground truth labels on the validation set, but calculate an unsupervised CIM loss on it, since we use this loss in the training process to guide the optimization direction.
>
> We have claimed the selection method in the experiment setup in Section 5.1 (L267), and we will make this clear in the method part for clarity.
>
>
> **Q5:** The linear programming (6) misses the condition $\sum w =1$? Does this affect the solution of (6), and consequently the performance of the DA framework?
>
> **R5:** Thanks for pointing this out. In fact we have imposed the constraints $w \in S^m = \\{w \in \mathbb{R}^m_+ | \sum^m_{j=1} w_j = 1\\}, m = 3$ when we first introduce $w$ (L203). And the $w \in S^m$ in the optimization objective of Eq. 6
> $
> w^∗ = \argmax_{w\in S^m} (Gw)^T (I(L_{Val} > 0)g_v + I(L_{Val} = 0)G\mathcal{1}/m)$
> also indicates the constraints on $w$.
>
> We will make this clearer in the final version.

---

> > ### Comment · Reviewer_MkjU · 2021-08-16
> > **Please also address my concerns about the quality of the paper**
> >
> > Thanks to the authors for your response and clarifications.  I would say most of my concerns are solved, except the two following ones:
> >
> > 1. Theoretical guarantee for the existence of a Pareto optimal of the corresponding multi-objective optimization problem by using the proposed ParetoDA.
> > 2. An upper bound for the generalization error of the DA method.
> >
> > I would appreciate and reconsider the paper score if the authors can address those remaining things.

---

> > > ### Author Response · Authors · 2021-08-16
> > > **Response to the concerns about the quality of the paper**
> > >
> > > Thanks for your recognition for our response. Actually, we have already addressed your concerns about the quality of the paper in the comment "Response to Reviewer MkjU (PART I)" below. For your convenience, here we present our response again. Please let us know if there are any further questions.
> > >
> > >
> > > ## Response to Reviewer MkjU (PART I)
> > >
> > > Thank you for your valuable comments, and for recognizing the contributions in our work. We will incorporate your suggestions in the revision.
> > > We address your concerns and questions below. Please let us know if there are any further questions.
> > >
> > > ------------------------------------------------------------------------Quality--------------------------------------------------------------------------\
> > > **Q1:** There is no theoretical guarantee for the existence of a Pareto optimal of the corresponding multi-objective optimization problem by using the proposed ParetoDA.
> > >
> > > **R1:**
> > > Thanks for your suggestion. Below we provide the theoretical guarantee for the existence of a Pareto optimal of the corresponding multi-objective optimization problem by using the proposed ParetoDA.
> > >
> > > Similar as in [1], we provide a theoretical guarantee for the existence of a Pareto critical (also named as local Pareto optimal in [2]) solution of our method, where a solution is called Pareto critical if no other solution in its neighborhood can dominate this solution. Specifically, we show by Theorem 2 below that our method will optimize until the solution is Pareto critical and satisfies $\mathcal{L}_{Val}=0$.
> > >
> > > > [1] Lin, X., Zhen, H. L., Li, Z., Zhang, Q. F., & Kwong, S. (2019). Pareto multi-task learning. Advances in neural information processing systems, 32, 12060-12070.\
> > > [2] Mahapatra, D., & Rajan, V. (2020, November). Multi-task learning with user preferences: Gradient descent with controlled ascent in pareto optimization. In International Conference on Machine Learning (pp. 6597-6607). PMLR.
> > >
> > > **Theorem 2.**
> > >
> > > Let $\hat{\boldsymbol{\theta}}$ be the final output model parameters of our method. Let $\boldsymbol{w}^*$ be the solution of the problem in Eq. (6) based on $\hat{\boldsymbol{\theta}}$, and $\boldsymbol{d}^*=\boldsymbol{G}\boldsymbol{w}^*$ be the resulted update direction. We have:
> > >
> > > - if $\hat{\boldsymbol{\theta}}$ is not Pareto critical, then $\boldsymbol{d}^*\neq\boldsymbol{0}\in\mathbb{R}^d$;
> > >
> > > - if $\hat{\boldsymbol{\theta}}$ is Pareto critical, we further consider two cases:
> > >
> > >     1) If $\mathcal{L}_{Val}=0$, then $\boldsymbol{d}^*=\boldsymbol{0}\in\mathbb{R}^d$ or $(\boldsymbol{d}^*)^T\boldsymbol{G}=\boldsymbol{0}\in\mathbb{R}^3$. In either case, the update is meaningless and will halt.
> > >
> > >     2) If $\mathcal{L}_{Val}>0$, assuming the gradient of $\mathcal{L}_T$
> > >
> > >      is consistent with the gradient of $\mathcal{L}_{Val}$, i.e., $\boldsymbol{g}_3^T\boldsymbol{g}_v>0$,
> > > then  $\boldsymbol{d}^*\neq\boldsymbol{0}\in\mathbb{R}^d$.
> > >
> > > Note that the assumption that $\boldsymbol{g}_3^T\boldsymbol{g}_v>0$ for a Pareto critical $\hat{\boldsymbol{\theta}}$ is reasonable, because when $\hat{\boldsymbol{\theta}}$ is Pareto critical, the model is usually well-trained, and $\boldsymbol{g}_3,\boldsymbol{g}_v$ are gradients of the same loss on i.i.d sampled two datasets from the target domain. Then it is reasonable to assume the gradients are consistent.
> > >
> > > We prove Theorem 2 in the following.
> > >
> > > **Proof.**
> > >
> > > - If $\hat{\boldsymbol{\theta}}$ is not Pareto critical, we suppose $\boldsymbol{d}^*=\boldsymbol{G}\boldsymbol{w}^*=\boldsymbol{0}\in\mathbb{R}^d$, where $\boldsymbol{w}_j^*>0$ for all $j\in$ {1,2,3}.
> > >
> > >     Since $\hat{\boldsymbol{\theta}}$ is not Pareto critical, there exists a $\boldsymbol{d}'\neq\boldsymbol{0}\in\mathbb{R}^d$
> > > that satisfies $  (\boldsymbol{d}')^T\boldsymbol{g}_j>0$, for some $j\in$ {1,2,3};
> > > and $  (\boldsymbol{d}')^T\boldsymbol{g}_i\geq0$, for other $i\in$ {1,2,3} and $i\neq j$.
> > >
> > >    Since $\boldsymbol{G}\boldsymbol{w}^*=\boldsymbol{0}$,
> > > then
> > > $\boldsymbol{w}_j^* \boldsymbol{g}_j$
> > >
> > >     $= - \sum_{i\neq j}\boldsymbol{w}_i^* \boldsymbol{g}_i$ .
> > >
> > >     Also, because $  (\boldsymbol{d}')^T\boldsymbol{g}_j>0$, then $  (\boldsymbol{d}')^T(\boldsymbol{w}_j^*\boldsymbol{g}_j)>0$
> > >
> > >     and
> > > $(\boldsymbol{d}')^T(\sum_{i\neq j}\boldsymbol{w}_i^*\boldsymbol{g}_i)<0$.
> > >
> > >     However, $(\boldsymbol{d}')^T(\sum_{i\neq j}\boldsymbol{w}_i^*\boldsymbol{g}_i)<0$ is not possible because
> > > $  (\boldsymbol{d}')^T\boldsymbol{g}_i\geq0$, for $i\in$ {1,2,3} and $i\neq j$
> > > and
> > > $\boldsymbol{w}_i^*>0$ for all $i\in$ {1,2,3},
> > >
> > >     which generates a contradiction for $\boldsymbol{d}^*=\boldsymbol{0}\in\mathbb{R}^d$.
> > >
> > > - If $\hat{\boldsymbol{\theta}}$ is Pareto critical, we consider two cases.
> > >
> > >     - If $\mathcal{L}_{Val}=0$, then from Theorem 1 we know that $\boldsymbol{d}^*$ becomes a descent direction, i.e.,
> > >   $  (\boldsymbol{d}^*)^T\boldsymbol{g}_j\geq 0, \ \forall j\in$ {1,2,3}. \
> > > Since $\hat{\boldsymbol{\theta}}$ is Pareto critical, then there does not exist $ j\in$ {1,2,3} such that $  (\boldsymbol{d}^*)^T\boldsymbol{g}_j> 0$. Then$ (\boldsymbol{d}^*)^T\boldsymbol{g}_j= 0, \ \forall j\in$ {1,2,3}. \
> > > Thus, we either have $\boldsymbol{d}^*=\boldsymbol{0}\in\mathbb{R}^d$ or just $(\boldsymbol{d}^*)^T\boldsymbol{G}=\boldsymbol{0}\in\mathbb{R}^3$.
> > >
> > >     - Consider the second case, $\mathcal{L}_{Val}>0$, then $\boldsymbol{g}_v\neq\boldsymbol{0}\in\mathbb{R}^d$.\
> > > Given the assumption that $\boldsymbol{g}_3^T\boldsymbol{g}_v>0$, then $\boldsymbol{g}_3\neq\boldsymbol{0}\in\mathbb{R}^d$.
> > > Then $\boldsymbol{w}'$ can be $[0,0,1]$ and $\boldsymbol{\gamma}'=(\boldsymbol{G}\boldsymbol{w}')^T\boldsymbol{g}_v=\boldsymbol{g}_3^T\boldsymbol{g}_v>0$.
> > >
> > >         Therefore, the maximized $\boldsymbol{\gamma}^*\geq \boldsymbol{\gamma}'>0$,
> > >
> > >         then from Theorem 1 we know that $(\boldsymbol{d}^*)^T\boldsymbol{g}_v>0$, which means $\boldsymbol{d}^*\neq\boldsymbol{0}\in\mathbb{R}^d$.
> > >
> > > In summary, we can obtain a Pareto critical solution with simple iterative gradient-based update rule $\theta_{t+1} = \theta_t + \eta d$. We will add the above theoretical guarantee to the supplementary materials.
> > >
> > > **Q2:** It would be much more interesting if the authors can develop an upper bound for the generalization error of the DA.
> > >
> > > **R2:**
> > > Thanks for your valuable comment. Actually, we have not developed a strict upper bound for the generalization error of the target domain in DA. However, **it can be proved that our ParetoDA can effectively limit the terms of the existing upper bound** proposed by Ben-David et al. [3]. We provide the theoretical derivation process below.
> > >
> > > In UDA, the theory proposed by Ben-David et al. [3] provides an upper bound for the generalization error on target samples, which is mainly composed of three terms: 1) the expected error on source domain $\epsilon_S(h)$; 2) the $H\Delta H$ divergence between source and target domain $d_{H\Delta H}(D_s, D_t)$; 3) the combined error $\lambda$ of the ideal joint hypothesis $h^*$. The formulation is as follows:
> > >
> > > $\epsilon_T(h) \le \epsilon_S(h) + 1/2 d_{H\Delta H}(D_s, D_t)+ \lambda $;
> > >
> > > where $\lambda = \epsilon_S(h^*) + \epsilon_T(h^*)$ and $h^* = \argmin_{h\in H} \epsilon_S(h)+\epsilon_T(h)$.
> > >
> > > The goal of the UDA model is to limit the upper bound of the expected error on the target samples to a small value.
> > > In previous DA methods, the first term, $\epsilon_S(h)$, is expected to be small because reliable labels are owned in the source domain. And the third term $\lambda$ is also generally considered sufficiently small. Thus, most DA methods mainly focus on reduce the second term $d_{H\Delta H}(D_s, D_t)$ by domain alignment.
> > >
> > > However, if one of the three terms goes down and another term goes up, then the model still can not effectively reduce the upper bound of $\epsilon_T(h)$. Excessive either source domain classification or domain alignment will cause damage to the other.
> > >
> > > In our ParetoDA, by using MOO framework and pareto optimization, **we can simultaneously optimize the source classification loss and the domain alignment loss, so as to make $\epsilon_S(h)$ and $d_{H\Delta H}(D_s, D_t)$ decrease synchronously**. That is, in ParetoDA, the $\epsilon_S(h)$ is further reduced under the premise that this operation does not increase $d_{H\Delta H}(D_s, D_t)$ and vice versa.
> > >
> > > Moreover, the training of our proposed CIM loss on target samples and the optimization guidance of the surrogate loss on validation set $L_{Val}$, both contribute to the classification ability of the model to the target domain, and then **reduce the combined error $\lambda$ of the ideal joint hypothesis** $h^*$.
> > >
> > > In this way, **the upper bound of $\epsilon_T(h)$ can be effectively reduced in our work**. The above discussion will be added into supplementary materials to verify the effectiveness of our method theoretically.
> > >
> > > It is worth noting that in addition to the theoretical support, **the effectiveness of our method has already been strongly supported in the aspect of logic and experiment**.
> > >
> > >
> > > > [3] Shai Ben-David, John Blitzer, Koby Crammer, Alex
> > > Kulesza, Fernando Pereira, and Jennifer Wortman Vaughan.
> > > A theory of learning from different domains. In Machine
> > > learning, volume 79, pages 151–175, 2010. 2

---

> > > > ### Comment · Reviewer_MkjU · 2021-08-17
> > > > **Upgraded my score to 6**
> > > >
> > > > I appreciate the detailed response provided by the authors. I suggest the authors adding those things to the main paper or to the supplementary materials. I upgraded my score to 6.

---

> > > > > ### Author Response · Authors · 2021-08-18
> > > > > **Thank You & Authors' Response**
> > > > >
> > > > > Thank you for reading our response and increasing the score!
> > > > >
> > > > > We will improve the theoretical contents about the existence of solution and the upper bound for generalization error. We will add these contents to the later revised manuscript. Moreover, we will update the manuscript according to the other comments from the reviewers.
> > > > >
> > > > > Thanks again for your valuable feedback!

---

### Official Review · Reviewer_rFU1 · 2021-07-16

**Rating:** 7
**Confidence:** 3

**Summary:**

This paper addresses the issue of previous domain adaptation optimization scheme, which learning objectives conflict with each other. The main contribution is to dynamically control the overall optimization direction so that no objective is harmed resulting out to boost the performance. To this end, it designs desired pareto optimal solution with surrogate objective function that mimics the target classification task called Pareto Domain Adaptation.

**Limitations And Societal Impact:**

Author well express the limitation and potential negative societal impact in supplemental material.

**Main Review:**

Originality:

The main originality of this paper seems to come from two methodologies, CIM loss and Pareto optimal solution.  For CIM loss, utilizing domain label to compute class-wise domain prediction to modify the target predictions is thought to be novel.  And then, it shows well comparison with other gradient-based scheme and proves the effectiveness and originality of dynamic pareto optimal solutions.

Quality:

Designing process of CIM loss and Pareto Optimal solution is well supported by theoretical analysis and final results.
However, the major curiosity that I have is relationship between CIM loss and pareto optimal solution. Ablation study in Table 5. doesn't guarantee that CIM loss is better solution for ParetoDA compared with basic IM loss. It would be more convincing if the author can provide IM + ParetoDA results to prove CIM as a better guidance for the proposed adaptation method.
I also wonder whether this framework can be applied to self-training based domain adaptation[1], specifically where the explicit domain alignment module doesn't exist.
-  [1]  Domain Adaptation for Semantic Segmentation via Class-Balanced Self-Training,  Yang et al.


Clarity:

This paper is well written and easy to follow. Specifically, figure in this paper well describes the intention of the author and proves the effectiveness of suggested model.


Significance:

This paper seems to show quite significant result with boosting improvement  and robustness  to the different tasks and backbone model.

**Time Spent Reviewing:**

5 hours

---

> ### Author Response · Authors · 2021-08-10
> **Response to Reviewer rFU1**
>
> Thank you for your valuable feedback, and for appreciating the motivation of our work and our technical and theoretical contributions. We will incorporate your suggestions in the revision.
> Please see below for our response to your comments and let us know if there are any further questions.
>
> ------------------------------------------------------------------------Quality--------------------------------------------------------------------------\
> **Q1:** It would be more convincing if the author can provide IM + ParetoDA results to prove CIM as a better guidance for the proposed adaptation method.
>
> **R1:** Thank you for your advice, we supplement the experiment of IM + ParetoDA on office31 as your suggestion, the results are as follows:
>
> |      |IM    |CIM   |EPO   |ParetoDA|Accuracy|
> |:----:|:----:|:----:|:----:|:----:|:----:|
> |DANN  |      |      |      |      | 82.2 |
> |DANN  |  [o] |      |      |      | 87.9 |
> |DANN  |      |  [o] |      |      | 88.4 |
> |DANN  |      |      |  [o] |      | 87.7 |
> |DANN  |      |  [o] |  [o] |      | 89.9 |
> |**DANN**  |  **[o]** |      |      | **[o]**  | **89.7** |
> |**DANN**  |      |  **[o]** |      | **[o]**  | **90.2** |
>
> It can be seen that the version with the guidance of CIM is superior to the one with the guidance of IM, which demonstrates the effectiveness of our proposed CIM loss.
>
> Intuitively, CIM exploits additional domain labels to enhance the accuracy of target predictions based on Bayesian theory, and then adopts an information maximization mechanism on the improved predictions to mimics target supervision loss, while IM only applies the information maximization on the vanilla target predictions. Thus, our proposed CIM performs better than IM and can act as a preferable guidance for optimization since it can better simulate the supervision loss of the target domain.
>
>
>
> **Q2:** Whether this framework can be applied to self-training based domain adaptation[1], specifically where the explicit domain alignment module doesn't exist.
>
> **R2:** Thanks for your interest and the answer is yes.
>
> Our method aims to handle the optimization conflict problem that exists among multiple training objectives in DA, which affects the model performance on target domain. Since the most classical optimize objectives in DA are source classification loss $L_S$ and domain alignment loss $L_A$, we mainly focus on the conflict between them in this paper. However, ParetoDA is not limited to the domain alignment loss, it can be applied to other loss functions that have conflict with $L_S$, e.g., implicit domain alignment loss, regularization loss, self-training loss, etc. Moreover, ParetoDA on more objectives such as three losses is also workable.
>
> For this self-training based DA method (CBST), we can directly apply ParetoDA into the step b) in Eq. 3 or 6 or 9 (trains the network model given the pseudo-labels selected in step a)), since the second self-training term may conflict with the first source supervised term. To prove this, we conduct experiments on CBST+ParetoDA. The result is as follows:
>
> |Case|mIoU|Road|Sidewalk|Build|Wall|Fence|Pole|Traffic Light|Traffic Sign|Veg.|Terrain|Sky|Person|Rider|Car|Truck|Bus|Train|Motor|Bike|
> |:---:|:---:|:---:|:---:|:---:|:---:|:---:|:---:|:---:|:---:|:---:|:---:|:---:|:---:|:---:|:---:|:---:|:---:|:---:|:---:|:---:|
> |Source|35.4|70.0|23.7|67.8|15.4|18.1|40.2|41.9|25.3|78.8|11.7|31.4|62.9|29.8|60.1|21.5|26.8|7.7|28.1|12.0||ST|41.5|88.0|20.4|80.4|25.5|19.7|41.3|42.6|20.2|86.0|3.5|64.6|65.4|25.4|83.3|31.7|44.3|0.6|13.4|3.7|
> |CBST|45.2|86.8|46.7|76.9|26.3|24.8|42.0|46.0|38.6|80.7|15.7|48.0|57.3|27.9|78.2|24.5|49.6|17.7|25.5|45.1|
> |CBST + ParetoDA |48.1|82.4|50.7|80.2|34.9|36.7|45.9|47.2|45.1|72.9|27.8|46.0|62.3|33.0|81.4|29.5|44.3|14.0|29.8|49.3|
>
> Due to the limit of time, we only conduct experiments on the task GTA5-cityscapes. We use mIoU (mean IoU of 19 classes) as the metric as CBST does. It is clear that ParetoDA is still effective in the self-training based DA method.

---

### Official Review · Reviewer_DGKA · 2021-07-19

**Rating:** 6
**Confidence:** 4

**Summary:**

While doing domain adaptation, we usually have a source classification loss and a domain alignment loss. However, these losses might compete with each other and we might reach a situation where one loss can’t be reduced without degrading the other one. We call that the Pareto front. For DA, we don’t know which loss is better than the other one. In this paper, they come up with an optimization strategy that helps us navigate to an ideal portion on the Pareto front. Essentially, they stop the optimization if one loss can’t be improved without degrading the other.

**Limitations And Societal Impact:**

Yes, they have.

**Main Review:**

- Originality: 1. Seems to be a modified version of EPO where the only difference is that this one is specifically designed for DA. This one also uses a custom surrogate loss for DA while EPO uses KL divergence. 2. They came up with their own equations for 3. They claim that Pareto optimization hadn’t really been used before for DA, but I found a paper (Locality Preserving Projection for Domain Adaptation with Multi-Objective Learning Le Shu, Tianyang Ma, Longin Jan Latecki) which did. Needs a comparison against that.

- Quality: Well thought out and precise experiments. Technically sound with well-supported claims. However, I am worried we might get stuck in local minima in some situations (especially in the beginning stages of training) with their optimization strategy (losses need to increase sometimes to get out of local minima, but their conditions don’t allow for it). Would appreciate it if they could tell us about their experience wrt that.

- Clarity: The theoretical section 4 is not very well written and could be polished and made simpler, and easier to understand. One thing, I still don’t understand how you determine a solution to be an ideal solution on the Pareto front  (I get that we can’t choose anything on the extreme ends, but where did you get the Expected Solution on Fig. 1 and the ParetoDA (ideal) in Fig. 3). Is that from the intersection of the preference vector from the surrogate loss with the Pareto boundary?

- Significance: Overall, it is a plug-and-play optimization technique for DA which can be easily used with already existing SOTA model architectures to get performance boosts. Could be a valuable tool for DA. Maybe they should talk from the perspective of autoML for DA.

AFTER THE DISCUSSION PERIOD: I would like to thank the authors and reviewers and AC. I think that enough information and clarification have been exchanged so that the major issues have been adequately addressed. One of them was also the comparison with previous work, which the authors have addressed very effectively. I would have given a higher score if fewer things had to be revised for the camera ready, but anyways, I am in favor of accepting the paper.

**Time Spent Reviewing:**

15

---

> ### Author Response · Authors · 2021-08-10
> **Response to Reviewer DGKA （PART 1）**
>
> Thank you for your valuable feedback, and your suggestions on the presentation. We will incorporate your suggestions in the revision.
> Below we address all your concerns. Please let us know if there are any further questions.
>
> -----------------------------------------------------------------------Originality-------------------------------------------------------------------------\
> **Q1:** Seems to be a modified version of EPO where the only difference is that this one is specifically designed for DA. This one also uses a custom surrogate loss for DA while EPO uses KL divergence.
>
> **R1:**
> We respectfully disagree with the above assessment of the reviewer. Our ParetoDA is very different from EPO whether in terms of insight or technical details. We sincerely hope that you can reconsider the originality of our method.
>
> First, we would like to highlight our technical contributions. We handle the optimization conflict problem in domain adaptation (DA) by casting DA into a multi-objective optimization scheme, and we dynamically adjust the preference between training objectives to find an ideal Pareto optimal solution.
> Specifically, we propose two novel mechanisms to handle the unique problems in DA:
> 1) a target-prediction enhancing mechanism for the lack of supervision in the target domain, in which the enhancement is implemented by leveraging domain labels and Bayesian theory;
> 2) a dynamic preference learning mechanism for the lack of prior information about the preference between the training objectives, e.g., source classification loss $L_S$ and domain alignment loss $L_A$.
>
> Then, for the second mechanism, we take EPO as our basis just to inherit its characteristics that won't damage any optimization objective. However, EPO relies on pre-determined preference vector to get corresponding Pareto optimal solution, which is undetermined and changeable in DA literature. So we further design a dynamic preference learning mechanism based on our target-prediction enhancing mechanism, which can guide the optimization direction to reach an ideal solution on the Pareto front.
>
> At last, the core idea of EPO is to optimize strictly along the pre-determined preference vector to accurately reach the specified point on the Pareto front, that is, an advanced version of a previous work PMTL. On the contrary, our key idea is to dynamically guide the optimization direction according to the situation in the training process, so as to automatically get an ideal Pareto optimal solution.
>
> In summary, whether in terms of insight or technical details, our method is very different from EPO.
>
>
> **Q2:** Comparision against "Locality Preserving Projection for Domain Adaptation with Multi-Objective Learning".
>
> **R2:**
> Thanks for pointing it out. We carefully read the paper "Locality Preserving Projection for Domain Adaptation with Multi-Objective Learning"(hereinafter referred to as LPPDA) and compare it with our ParetoDA. The main differences are as follows:
>
> - **motivation**: LPPDA only focuses on the dual objective joint optimization of source supervised learning and target unsupervised learning, while does not notice the problem of domain alignment. By contrast, the key motivation of our ParetoDA is to handle the optimization conflict between source supervised learning and domain alignment learning, so as to improve the model performance on the target domain.
> - **insight**: LPPDA simply thinks that the two objectives of source supervised learning and target unsupervised learning contribute equally to the final performance. Thus, they think that any solutions on the Pareto front are equally good, ignoring the difference among solutions with different preferences on the results. They finally obtain a set of solutions. Instead, the core insight of ParetoDA is that the importance of multiple training objectives is unequal, undetermined and dynamic, which needs to be dynamically adjusted according to the guidance loss on the validation set, so as to attain a single, ideal Pareto optimal solution that performs best on the target domain.
> - **strategy**: LPPDA utilizes SVD decomposition to approximate the Pareto frontier when solves the Pareto optimization, which will cause an error. We have no approximation operation in ParetoDA.
> - **application**: LPPDA studies linear models and solves linear regression problems. While ParetoDA is suitable for arbitrary deep models, which can be applied to classification and segmentation problems and can be further extended to regression problems. Moreover, The Pareto optimization method proposed in LPPDA is specific to its method. While the dynamic Pareto optimization mechanism proposed by our ParetoDA can be applied to existing DA methods, which can improve their model performance significantly by changing their optimization scheme.
>
> Moreover, to further verify the superiority of our method, we reproduce the LPPDA and conduct experiments on office-home dataset, and then make a comparision with our ParetoDA. It can be seen that our method is far superior to LPPDA.
>
> Method |Ar:Cl |Ar:Pr| Ar:Rw| Cl:Ar |Cl:Pr| Cl:Rw| Pr:Ar |Pr:Cl| Pr:Rw |Rw:Ar| Rw:Cl |Rw:Pr| Avg.
> ---|---|---|---|---|---|---|---|---|---|---|---|--- | ---|
> LPPDA          |52.8| 69.4|73.5|58.6|70.7|70.5|60.2|53.5|77.6|70.1|57.4|81.2|66.3
> DANN + ParetoDA|55.2|74.4|79.0|61.9|72.4|72.9|62.1|55.8|81.1|74.4|61.1|82.4|69.4
> CDAN + ParetoDA|56.8|75.9|80.5|64.4|73.5|73.7|65.6|55.2|81.3|75.2|61.1|83.9|70.6
>
> ------------------------------------------------------------------------Quality--------------------------------------------------------------------------\
> **Q3:** Worried we might get stuck in local minima in some situations with our optimization strategy.
>
> **R3:**
> Thank you for your valuable comments, but in fact, our optimization strategy allows losses to increase sometimes to get out of local minima.
>
> In our optimization strategy, there exist two modes according to different cases. One is **pure descent mode**, in which the update direction can decrease all the training losses simultaneously; the other is **guidance descent mode**, in which the update direction is forced to decrease the loss whose gradient is the most consistent with the guidance gradient $g_v$ while the losses that have negative angles with $g_v$ are allowed to ascend. Thus, our optimization strategy is able to jump out of local minima for achieving better results on the validation set, which indicates better performance on the target domain.
>
> Please see L228-248 in our paper for details.
>
>
> ------------------------------------------------------------------------Clarity--------------------------------------------------------------------------\
> **Q4:** The theoretical section 4 is not very well written and could be polished and made simpler, and easier to understand.
>
> **R4:**
> Thanks for pointing this out. We will optimize the presentation of our theories and formulations in Section 4 to make them easier to understand as your suggestion, e.g., simplify the formula 3, put some unnecessary derivation processes into the supplementary materials, add some explanations to make the theory clearer.
>
> **Q5:** Elaboration about how to determine a solution to be an ideal solution on the Pareto front.
>
> **R5:**
> Different from the standard MOO problem, the ultimate goal of DA is to improve the classification performance of the model on target domain by jointly reducing the source classification loss $L_S$ and the domain alignment loss $L_A$. Obviously, the ideal Pareto optimal solution is the one that can minimize the target classification loss.
>
> To show the ideal solution on the Pareto front, we utilize the gradient of supervised classification loss (using ground-truth label) on the validation set (split from target domain) to guide the optimization direction, so as to find a Pareto optimal solution that is most favorable for the model performance on the validation set, which also implies better performance on the target test set. That's how we get **the ideal Pareto solution in Fig. 3**. Note that it won't be overfitted to the validation set since our optimization strategy also requires that the next solution $\theta_{t+1}$ is not dominated by the current solution $\theta_t$.
>
> However, the ground-truth label of the target domain is unavailable. Therefore, we propose a target-prediction enhancing mechanism and design a surrogate loss CIM based on it, which can substitute for the supervised classification loss on the validation set to dynamically guide the optimization direction. By following its gradient descending direction, we can get a relatively **expected Pareto optimal solution**. It can be seen that ParetoDA reaches the Pareto optimal solution closest to the ideal one in Fig. 3, which also verifies the effectiveness of our proposed surrogate loss.
>
> As for **the expected solution in Fig. 1**, it is only a schematic point.
>
> We will clarify this strategy more clearly in the introduction.

---

> ### Author Response · Authors · 2021-08-19
> **Response to Reviewer DGKA （PART 2）**
>
> -----------------------------------------------------------------------Significance-------------------------------------------------------------------------\
> **Q6:** Maybe they should talk from the perspective of autoML for DA.
>
> **R6:**
>
> Thanks for your insightful suggestion.
>
> Automated machine learning (AutoML) aims to set manual efforts free from tedious tasks, such as tuning machine learning models, model selection, feature engineering, hyperparameter optimization, nerual architecture search and so on.
>
> Take Neural Architecture Search (NAS) as an example, the goal of NAS is to automatically search the network architecture appropriate for various application task, such as object recognition and semantic segmentation. Similarly, our ParetoDA attempts to automatically search a suitable configuration of weight hyperparameters tailed for Domain Adaptation (DA). **From this point of view, our method can indeed be explained from the perspective of autoML for DA.**
>
> In the context of DA, the commonly used manual weight hyperparameter optimization methods (e.g., random search, grid search) are time consuming and unreliable due to the lack of target supervision. Moreover, the optimal weight hyperparameters for training objectives might be varing during the training process, which cannot be achieved by manual selection strategy.
>
> To address the questions, our ParetoDA proposes to automatically optimize the weight hyperparameters with the guidance of the gradient of a novel CIM loss, which can mimic the target supervision. Thus, we can search the solution desirable for the model performance on target classification automatically. The superiority of ParetoDA has been demostrated both theoretically and experimentally.
>
> We will take your valuable suggestion and add the discussion about ParetoDA from the perspective of autoML in the later manuscript.

---

> ### Author Response · Authors · 2021-08-27
> **Dear reviewer**
>
> Dear reviewer:
>
> Thanks a lot for your efforts in reviewing this paper. We have tried our best to address all your concerns and provided clarifications on all confusing concepts. Please let us know whether there are any unclear explanations. In addition, if you have any further questions, we will also be very glad to further clarify them.
>
> Sincerely thanks,
>
> Authors

---

### Decision · Program_Chairs · 2021-09-27

**Decision:**

Accept (Poster)

**Comment:**

The paper proposes a novel way to solve the Domain adaptation problem by formulating it as a multi-objective optimization problem instead of a sum of objective values. This allows for solving the DA problem with less hyper parameters that are hard to select in practice and can have an important impact fr applications of DA. Despite this original idea the paper had originally  borderline reviews due to some missing explanations and experiments. The authors did a very good job at answering the reviewers concerns with new experiments and even theoretical results. The reviewers agreed after the replies  that the paper is interesting and deserve to be published.

Please take into account all reviewers comments (missing references and clarifications) and insert the novel results (both numerical and theoretical) in the main paper and supplementary for the final version because the consensus among reviewers is that it will make the paper much better and interesting for the community.